# Efficient Reinforcement Learning by Guiding World Models with Non-Curated Data

**Yi Zhao**[†,1] **Aidan Scannell**[1,2] **Wenshuai Zhao**[1,3] **Yuxin Hou**[4] **Tianyu Cui**[1,5]
**Le Chen**[6] **Dieter Büchler**[6,7,8,9] **Arno Solin**[1,3] **Juho Kannala**[1,10] **Joni Pajarinen**[1]
[1]Aalto University [2]University of Edinburgh [3]ELLIS Institute Finland [4]Deep Render
[5]Imperial College London [6]Max Planck Institute for Intelligent Systems [7]CIFAR AI Chair
[8]University of Alberta [9]Alberta Machine Intelligence Institute (Amii) [10]University of Oulu

## Abstract

Leveraging offline data is a promising way to improve the sample efficiency of online reinforcement learning (RL). This paper expands the pool of usable data for offline-to-online RL by leveraging abundant non-curated data that is reward-free, of mixed quality, and collected across multiple embodiments. Although learning a world model appears promising for utilizing such data, we find that naive fine-tuning fails to accelerate RL training on many tasks. Through careful investigation, we attribute this failure to the distributional shift between offline and online data during fine-tuning. To address this issue and effectively use the offline data, we propose two techniques: *i)* experience rehearsal and *ii)* execution guidance. With these modifications, the non-curated offline data substantially improves RL's sample efficiency. Under limited sample budgets, our method achieves nearly twice the aggregate score of learning-from-scratch baselines across 72 visuomotor tasks spanning 6 embodiments. On challenging tasks such as locomotion and robotic manipulation, it outperforms prior methods that utilize offline data by a decent margin.

## 1 Introduction

Leveraging offline data offers a promising way to improve the sample efficiency of reinforcement learning (RL). Prior work has focused primarily on utilizing curated offline data labeled with rewards (Levine et al., 2020; Kumar et al., 2020; Fujimoto & Gu, 2021; Kumar et al., 2023), which is expensive and laborious to obtain. For instance, leveraging offline datasets for new robotic manipulation tasks requires retrospectively annotating image-based data with rewards. We instead propose expanding the pool of usable offline data by utilizing abundant non-curated data that is reward-free, of mixed quality, and collected across multiple embodiments. This leads to our main research question:

*How can we effectively leverage non-curated offline data for efficient RL?*

Typical offline-to-online RL methods (Lee et al., 2022; Zhao et al., 2022; Yu & Zhang, 2023; Nakamoto et al., 2024; Nair et al., 2020) fail to utilize non-curated offline data due to their assumption of structured data with rewards. While pre-training visual encoders (Schwarzer et al., 2021; Nair et al., 2022; Parisi et al., 2022; Xiao et al., 2022; Yang & Nachum, 2021; Shang et al., 2024) is a common approach to utilize non-curated offline datasets, it fails to fully leverage the rich information, such as dynamics models, informative states, and action priors for policy learning. On the other hand, learning world models from offline data appears promising for utilizing the non-curated dataset. However, prior work has explored world model training primarily in settings with known rewards (Lu et al., 2023; Rafailov et al., 2023; Hansen et al., 2024) or expert demonstrations (Zhu et al., 2024a; Zhou et al., 2024; Gao et al., 2025) or focused solely on visual prediction (Zheng et al., 2024; Zhu et al., 2024b). Recent approaches (Seo et al., 2022; Wu et al., 2024; 2025) have developed novel architectures for world model pre-training using in-the-wild action-free data, but paid limited

---

[†] Correspondence to `yi.zhao@aalto.fi`. Code and datasets: https://github.com/zhaoyi11/ncrl.

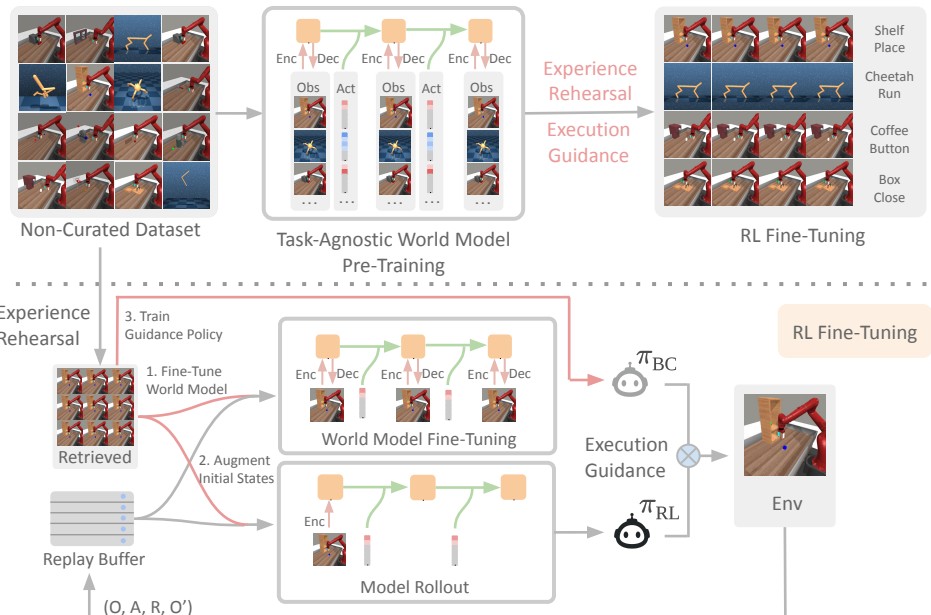

Figure 1: **Overview of NCRL (Non-curated offline data for efficient RL).** NCRL leverages non-curated offline data—reward-free, mixed-quality, and multi-embodiment—to enable efficient RL. It uses this data to pretrain a task-agnostic world model, and then, during fine-tuning, to reduce distributional shift and guide exploration through experience rehearsal and execution guidance.

attention to the fine-tuning process. As a result, despite being trained on massive datasets, these methods show only marginal improvements over training-from-scratch baselines. Additionally, due to the computational costs of RL experiments, previous work (Wu et al., 2024; 2025) evaluated only on a small set of tasks, leaving the effectiveness of the learned world model unclear on broader tasks. In contrast, we extensively evaluate our method on 72 visuomotor control tasks spanning both locomotion and robotic manipulation, demonstrating consistent improvements over existing approaches.

Through experiments, we observe that naively fine-tuning a world model fails to improve RL's sample efficiency on many tasks. With careful investigation, we identify the root cause as a distributional shift between offline data used for pre-training and online data used for RL fine-tuning. Specifically, when the offline data distribution does not sufficiently cover the data distribution of downstream tasks, the pre-trained world models struggle to benefit policy learning due to this distribution mismatch, shown in Fig. 2. Building on these insights, we propose using non-curated offline data in *both* pre-training and fine-tuning stages, in contrast to previous methods that only consider the offline data for world model pre-training (Wu et al., 2025; Yuan et al., 2022; Rajeswar et al., 2023). To this end, we propose a pipeline named Non-curated offline data for efficient RL (NCRL). In the pre-training stage, NCRL learns a task-agnostic world model from non-curated offline data that is reward-free, mix-quality and task-agnostic. In the fine-tuning stage, NCRL reuses this data through *experience rehearsal* and *execution guidance* to mitigate distributional shift by retrieving task-relevant trajectories and to prompt exploration by steering the agent toward regions where the world model has high confidence.

Equipped with our proposed techniques, NCRL demonstrates strong performance across a diverse set of tasks. Specifically, under a limited sample budget (150k samples), NCRL achieves almost double the aggregate score of learning-from-scratch baselines (DrQ-v2 and DreamerV3), while matching their performance achieved with larger sample budgets. On representative challenging tasks, NCRL outperforms baselines that leverage offline data as well as state-of-the-art methods using pre-trained world models by a significant margin. Additionally, without any modifications, we show that NCRL improves task adaptation, enabling agents to efficiently adapt their skills to new tasks. To summarize, our contributions are:

**C1** We propose a more realistic setting for leveraging offline data that consists of reward-free and mixed-quality multi-embodiment data.

Table 1: Comparison with different policy learning methods that leverage offline data.

| | Offline RL | Off2On RL | RLPD | MT Offline RL | NCRL (ours) |
|---|---|---|---|---|---|
| Reward-free offline data | ✗ | ✗ | ✗ | ✗ | ✓ |
| Non-expert offline data | ✓ | ✓ | ✓ | ✓ | ✓ |
| X-embodiment offline data | ✗ | ✗ | ✗ | ✓ | ✓ |
| Continual improvement | ✗ | ✓ | ✓ | ✗ | ✓ |
| Training stability | ✗ | ✗ | ✓ | ✗ | ✓ |

**C2** We demonstrate that naive world model fine-tuning fails on many tasks due to distributional shift between pre-training and fine-tuning data.

**C3** We propose two techniques, experience rehearsal and execution guidance, to mitigate the distributional gap and encourage exploration during RL fine-tuning.

**C4** We present NCRL, which leverages non-curated offline data in both pre-training and fine-tuning stages and clearly outperforms existing approaches across a diverse set of tasks.

## 2 RELATED WORK

In this section, we review RL methods that leverage offline data in different ways. See Sec. C for extended discussion and Table 1 for a comparison.

**RL with task-specific offline datasets** Offline RL trains agents purely from offline data by constraining divergence from behavior policies (Kumar et al., 2020; Fujimoto & Gu, 2021; Kumar et al., 2019; Wu et al., 2019; Kostrikov et al., 2021; 2022; Uchendu et al., 2023), but performance depends heavily on dataset quality (Yarats et al., 2022). Offline-to-online RL (Lee et al., 2022; Zhao et al., 2022; Yu & Zhang, 2023; Nair et al., 2020; Rafailov et al., 2023) addresses this by fine-tuning the agent via interaction with the environment. MOTO (Rafailov et al., 2023) proposes a model-based offline-to-online RL method with reward-labeled data, but requires model-based value expansion, policy regularization, and controlling epistemic uncertainty to conduct stable online RL training. Recent work (Ball et al., 2023; Li et al., 2023) demonstrates promising results by leveraging offline data, but it still assumes reward-labeled offline data or relies on near-expert data of the target tasks (Li et al., 2023), while we focus on a more general setting assuming reward-free, mixed-quality and task-agnostic offline data.

**RL with multi-task offline datasets** Recent work has explored multi-task offline RL (Kumar et al., 2023; Hansen et al., 2024; Julian et al., 2020; Kalashnikov et al., 2021; Yu et al., 2021), but requires known rewards. PWM (Georgiev et al., 2024) trains a world model for multi-task RL but is limited to state-based inputs and reward-labeled data. To handle unknown rewards, approaches like human labeling (Cabi et al., 2020; Singh et al., 2019), inverse RL (Ng et al., 2000; Abbeel & Ng, 2004), or generative adversarial imitation learning (Ho & Ermon, 2016) can be used, though these require human labor or expert demonstrations. Yu et al. (2022) assigns zero rewards to unlabeled data, which introduces additional bias. Apart from these, there is a line of work that focuses on representation learning or dynamics model training from in-the-wild data (Schwarzer et al., 2021; Parisi et al., 2022; Yang & Nachum, 2021; Yuan et al., 2022; Stooke et al., 2021; Shah & Kumar, 2021; Wang et al., 2022; Sun et al., 2023; Ze et al., 2023; Ghosh et al., 2023; Wu et al., 2025; 2023) but fails to utilize rich information in the dataset at the fine-tuning stage.

## 3 METHODS

In this section, we detail our two-stage approach, which consists of *(i)* world model pre-training, which learns a multi-task & embodiment world model, given offline data, which rather importantly, includes reward-free and mixed-quality data, and *(ii)* RL-based fine-tuning which leverages the pre-trained world model, non-curated offline data, and online interaction in an offline-to-online fashion. See Fig. 1 for the overview and Alg. 1 for the full algorithm.

## 3.1 PROBLEM SETUP

In this paper, we assume the agent has access to a non-curated but in-domain offline dataset $\mathcal{D}_{\text{off}}$ with three key characteristics: *(i)* trajectories lack reward labels $r_t^i$, *(ii)* data quality is mixed, and *(iii)* data comes from multiple embodiments. During fine-tuning, the agent interacts with the environment to collect labeled trajectories $\tau_{\text{on}}^i = \{o_t^i, a_t^i, r_t^i\}_{t=1}^T$ and stores them in an online dataset $\mathcal{D}_{\text{on}} = \{\tau_{\text{on}}^i\}_{i=1}^{N_{\text{on}}}$. Our goal is to learn a high-performance policy by leveraging both $\mathcal{D}_{\text{off}}$ and $\mathcal{D}_{\text{on}}$ while minimizing the required online interactions $N_{\text{on}}$.

## 3.2 MULTI-EMBODIMENT WORLD MODEL PRE-TRAINING

During pre-training, rather than training separate models per task as in previous work (Hafner et al., 2020; 2021; 2023), we train one world model per benchmark and demonstrate that a single multi-task & embodiment world model can effectively leverage non-curated data.

Since our primary goal is enabling RL agents to use non-curated offline data rather than proposing a new architecture, we adopt the widely-used recurrent state space model (RSSM) (Hafner et al., 2019) with several modifications: *(i)* removal of task-related losses, *(ii)* zero-padding of actions to unify dimensions across embodiments, and *(iii)* scaling the model to 280M parameters. With these changes, we show that RSSMs can successfully learn the dynamics of multiple embodiments and can be fine-tuned for various tasks.

Our first stage pre-trains the following components:

$$\text{Sequence model}: h_t = f_\theta(h_{t-1}, z_{t-1}, a_{t-1}) \qquad \text{Encoder}: z_t \sim q_\theta(z_t \,|\, h_t, o_t)$$
$$\text{Dynamics predictor}: \hat{z}_t \sim p_\theta(\hat{z}_t \,|\, h_t) \qquad \text{Decoder}: \hat{o}_t \sim d_\theta(\hat{o}_t \,|\, h_t, z_t).$$

The models $f_\theta$, $q_\theta$, $p_\theta$ and $d_\theta$ are jointly optimized by minimizing:

$$\mathcal{L}(\theta) = \mathbb{E}_{(o_{t-1}, a_{t-1}, o_t) \sim \mathcal{D}_{\text{off}}, z_t \sim q_\theta(\cdot \,|\, h_t, o_t)} \left[ \frac{1}{T} \sum_{t=1}^T \left( \beta_1 \mathcal{L}_{\text{pred}}(\theta) + \beta_2 \mathcal{L}_{\text{dyn}}(\theta) + \beta_3 \mathcal{L}_{\text{rep}}(\theta) \right) \right], \quad (1)$$

where $\beta_1, \beta_2, \beta_3$ are weights of each term. $\mathcal{L}_{\text{pred}}$ minimizes reconstruction error, $\mathcal{L}_{\text{dyn}}$ enables the sequence model and dynamics predictor to predict future latent states, and $\mathcal{L}_{\text{rep}}$ encourages the representation to be more predictable. They are given as:

$$\begin{aligned}
\mathcal{L}_{\text{pred}}(\theta) &= -\ln d_\theta(o_t \,|\, z_t, h_t) \\
\mathcal{L}_{\text{dyn}}(\theta) &= \max\left(1, \text{KL}\big(\text{sg}(q_\theta(z_t \,|\, h_t, o_t) \| p_\theta(\hat{z}_t \,|\, h_t))\big)\right) \\
\mathcal{L}_{\text{rep}}(\theta) &= \max\left(1, \text{KL}\big(q_\theta(z_t \,|\, h_t, o_t) \| \text{sg}(p_\theta(\hat{z}_t \,|\, h_t)))\big)\right),
\end{aligned} \quad (2)$$

where sg represents the stop-gradient operator and $\text{KL}(p \| q)$ is the KL divergence.

While there is room to improve world model pre-training through recent self-supervised methods (Eysenbach et al., 2023) or advanced architectures (Vaswani et al., 2017; Gu et al., 2022; Mereu et al., 2025), such improvements are orthogonal to our method and left for future work.

## 3.3 RL-BASED FINE-TUNING WITH REHEARSAL AND GUIDANCE

In our fine-tuning stage, the agent interacts with the environment to collect new data $\tau_{\text{on}}^i = \{o_t^i, a_t^i, r_t^i\}_{t=0}^T$. This data is used to learn a reward function $\hat{r}_t \sim r_\theta(\hat{r}_t \,|\, h_t, z_t)$ via supervised learning while fine-tuning the world model with Eq. (1). For simplicity, we denote the concatenation of $h_t$ and $z_t$ as $s_t = [h_t, z_t]$ and use $\hat{s}_t = [h_t, \hat{z}_t]$ when the latent state is predicted by the dynamics predictor $p_\theta$. The actor and critic are trained using imagined trajectories $\hat{\tau}^i = \{\hat{s}_t^i, a_t^i\}_{t=0}^T$ generated by rolling out the policy $\pi_\phi(a \,|\, s)$ with the sequence model $f_\theta$ and the dynamics predictor $p_\theta$. The rollouts are initialized from states $p_0(s)$ sampled from the replay buffer. The critic $v_\phi(V_t^\lambda \,|\, s_t)$ learns to approximate the distribution over the $\lambda$-return $V_t^\lambda$, calculated as:

$$\underbrace{V_t^\lambda}_{\lambda-\text{return}} = \hat{r}_t + \gamma \begin{cases} (1-\lambda)v_{t+1}^\lambda + \lambda V_{t+1}^\lambda & \text{if } t < H \\ v_H^\lambda & \text{if } t = H \end{cases} \quad (3)$$

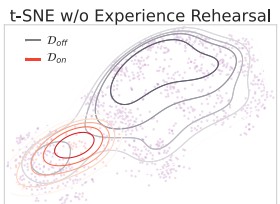 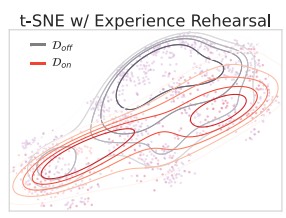 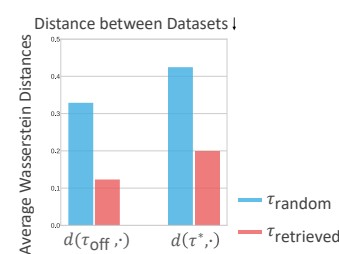

Figure 2: **Visualization of Distribution Mismatch. Left:** At the early stage of fine-tuning, there is a distribution shift between offline data used for world model pre-training and online data used for RL fine-tuning, which hurts performance. **Middle:** Experience rehearsal mitigates the distributional shift issue. **Right:** Quantitatively, at the early stage of fine-tuning, experience rehearsal reduces the Wasserstein distance between the online data and both the offline and expert data.

where $v_t^\lambda = \mathbb{E}[v_\phi(\cdot \,|\, s_t)]$ denotes the expectation of the value distribution predicted by the critic. The value function $v_\phi$ is trained by maximizing the log likelihood of the target $\lambda$-return, while the actor $\pi_\phi$ is optimized to maximize the $\lambda$-return by backpropagating gradients through the actions and latent states of the imagined trajectories:

$$\mathcal{L}(v_\phi) = \mathbb{E}_{p_\theta, \pi_\phi}\left[-\sum_{t=1}^{H-1} \ln v_\phi(V_t^\lambda \,|\, s_t)\right], \quad \mathcal{L}(\pi_\phi) = \mathbb{E}_{p_\theta, \pi_\phi}\left[\sum_{t=1}^{H-1}\left(-V_t^\lambda - \eta \cdot \mathbf{H}[a_t \,|\, s_t]\right)\right]. \quad (4)$$

For further details, we refer to DreamerV3 (Hafner et al., 2023)[1].

**Why Fine-Tuning a World Model Alone is Not Enough?** While previous methods typically discard non-curated offline data during fine-tuning (Wu et al., 2025; Rajeswar et al., 2023; Wu et al., 2023), we find that relying solely on a pre-trained world model often fails, particularly on hard-exploration tasks. To understand why, we analyze the Shelf Place task from Meta-World (Yu et al., 2020) as an illustrative task by visualizing the distributions of offline data $\mathcal{D}_{\text{off}}$ used for world model pre-training and online data $\mathcal{D}_{\text{on}}$ collected during early RL training in Fig. 2. The t-SNE plot in Fig. 2 (left) reveals a distribution mismatch between $\mathcal{D}_{\text{off}}$ and $\mathcal{D}_{\text{on}}$, leading to three key issues: *(i)* The world model's accuracy can degrade if a significant distributional shift exists between the offline and online data. This degradation is particularly pronounced when the offline data distribution is narrow, which may create a substantial state-space gap. *(ii)* For hard exploration tasks, the agent struggles to reach high-reward regions, causing the world model to be fine-tuned on a narrow online data distribution and leading to catastrophic forgetting. *(iii)* The policy update in Eq. (4) relies on imagined trajectories $\tilde{\tau} = p_0(s)\prod_{t=0}^{H-1}\pi_\phi(a_t \,|\, s_t)p_\theta(s_{t+1} \,|\, s_t, a_t)$, where $p_0(s)$ is sampled from $\mathcal{D}_{\text{on}}$. A narrow $p_0(s)$ limits the world model to rollout promising trajectories for policy updates. To address these challenges, we introduce two key components: *i)* experience rehearsal, which mitigates distributional shift by retrieving task-relevant trajectories from non-curated datasets (Fig. 2 middle, right), and *ii)* execution guidance, which encourages exploration by steering the agent toward regions where the world model has high confidence.

**Experience Rehearsal** Prior works like RLPD (Ball et al., 2023) and ExPLORe (Li et al., 2023) have shown that replaying offline data can boost RL training. However, these methods use small, well-structured offline datasets. In our setting, directly replaying non-curated offline data is infeasible since our datasets are ∼100× larger and contain diverse tasks and embodiments.

We propose retrieving task-relevant trajectories $\mathcal{D}_{\text{retrieved}} = \{\tau_{\text{retrieved}}^i\}_{i=1}^N$ from the non-curated offline data based on neural feature distance between online samples and offline trajectories. This filters out irrelevant trajectories, creating a small task-relevant dataset. Specifically, we compute:

$$\mathbf{D} = \|e_\theta(o_{\text{on}}) - e_\theta(o_{\text{off}})\|_2, \quad (5)$$

where $e_\theta$ is the encoder learned during world model pre-training, and $o_{\text{on}}$ and $o_{\text{off}}$ are initial observations from trajectories in the online buffer and offline dataset, respectively. For efficient search to get the top-k similar trajectories, we pre-compute key-value pairs mapping trajectory IDs to neural

---

[1]We follow the policy update described in the first version of the paper (https://arxiv.org/abs/2301.04104v1) and Dreamer v2 (Hafner et al., 2021).

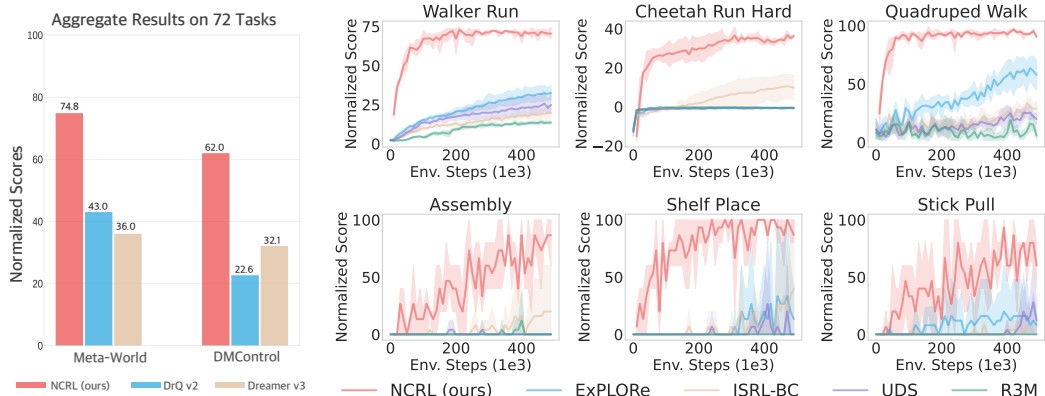

Figure 3: **Left:** Quantitative comparison across 72 diverse tasks from Meta-World (Yu et al., 2020) and DMControl (Tassa et al., 2018) with the same sample budget (150k). See Sec. I for full results. **Right:** Learning curves on representative challenging locomotion and robotic manipulation tasks. NCRL consistently outperforms state-of-the-art methods that leverage offline data by a decent margin. We plot the mean and corresponding 95% confidence interval.

features and use Faiss (Douze et al., 2024), enabling retrieval in seconds. The retrieval precision can be found in Sec. A.3.

The retrieved data is replayed during fine-tuning, so-called experience rehearsal. The retrieved data serves three purposes, as shown in Fig. 1. First, it prevents catastrophic forgetting by continuing to train the world model on relevant pre-training data, particularly important for hard exploration tasks with narrow online data distributions. Second, it augments the initial state distribution $p_0(s)$ during model rollout, enabling the world model to rollout promising trajectories for policy learning. Third, as described below, it enables learning a policy prior for execution guidance. In Sec. B.1, we explain that experience retrieval reduces distribution shift during online fine-tuning. We further demonstrate that experience retrieval acts as a regularizer, helping to prevent catastrophic forgetting during the fine-tuning process. Unlike RLPD and ExPLORe, we do not use this data to learn a Q-function, eliminating the need for reward labels.

**Execution Guidance via Prior Actors** Standard RL training initializes the replay buffer with random actions and collects new data through environment interaction using the training policy. However, offline data often contains valuable information like near-expert trajectories and diverse state-action coverage that should be utilized during fine-tuning. Additionally, distribution shift between offline and online data can degrade pre-trained model weights, making it important to guide the online data collection toward the offline distribution at the early training stage.

To achieve this, we train a prior policy $\pi_{bc}$ via behavioral cloning on the retrieved offline data $\mathcal{D}_{retrieved}$. During online data collection, we alternate between this prior policy $\pi_{bc}$ and the RL policy $\pi_\phi$ according to a pre-defined schedule. Specifically, at the start of each episode, we probabilistically select whether to use $\pi_{bc}$. If $\pi_{bc}$ is selected, we randomly choose a starting timestep $t_{bc}$ and duration $H$ during which $\pi_{bc}$ is active, with $\pi_\phi$ used for the remaining timesteps. In Sec. B.2, we theoretically show that, assuming $\pi_{bc}$ outperforms $\pi_\phi$ in the early stage of training, using a mixed policy composed of $\pi_{bc}$ and $\pi_\phi$ leads to improved policy performance.

While this approach shares similarities with JSRL (Uchendu et al., 2023), our method differs in three key aspects: *i)* we leverage non-curated rather than task-specific offline data, *ii)* we demonstrate the benefits of a model-based approach over JSRL's model-free framework, and *iii)* we randomly switch between policies mid-episode rather than only using $\pi_{bc}$ at episode start. The complete algorithm and theoretical analysis can be found in Sec. H and Sec. B, respectively.

## 4 EXPERIMENTS

In the experiments, we aim to answer the following questions:

**Q1** How does NCRL compare to state-of-the-art methods that leverage offline data and train-from-scratch baselines in terms of sample efficiency and final performance?

**Q2** How does NCRL compare to other leading model-based approaches that utilize offline data?

**Q3** How effectively does NCRL adapt to new tasks in a continual learning setting? We further conduct detailed ablation studies to evaluate our method.

**Tasks** We evaluate our method on *pixel*-based continuous control tasks from DMControl (Tassa et al., 2018) and Meta-World (Yu et al., 2020). The chosen tasks include both locomotion and manipulation tasks covering different challenges in RL, including high-dimensional observations, hard exploration, and complex dynamics. We use three random seeds for each task.

**Dataset** Our dataset consists of data from two benchmarks: DMControl and Meta-World, visualized in Sec. K. For DMControl, we include 10k trajectories covering 5 embodiments collected by *unsupervised RL agents* (Rajeswar et al., 2023; Pathak et al., 2017), trained via curiosity without task-related information. These trajectories vary in competence and coverage. As the unsupervised RL agents are trained to maximize the agent's curiosity rather than a specific reward signal, the dataset for DMControl does not contain expert trajectories for a specific task (e.g., Walk, Run etc.) For Meta-World, we collect mixed-quality 50k trajectories across 50 tasks using TDMPC-v2 agents (Hansen et al., 2024) by injecting Gaussian noise with $\sigma$ up to 2.0, which intentionally corrupts the policies and produces trajectories of varying success and quality. In practice, such a mixture of successful, partially successful, and failed behaviors can naturally arise from, for instance, noisy or partial human demonstrations collected through teleoperation. In Fig. 9, we assess the dataset quality via imitation learning, showing unsatisfactory performance. This emphasizes the mixed-quality property of the dataset. When combined with the DMControl data, our complete offline dataset comprises 60k trajectories (10M state-action pairs) across 6 embodiments.

### 4.1 NCRL Improves Sample Efficiency Across Diverse Tasks

**Comparison with Methods that Leverage Offline Data** We compare NCRL against several state-of-the-art methods that leverage reward-free data to improve RL training: *(i)* **R3M** (Nair et al., 2022), a visual representation pre-training approach that serves as our baseline for comparing pre-trained visual features using non-curated offline data. *(ii)* **UDS-RLPD** (Yu et al., 2022; Ball et al., 2023), which assigns zero rewards to offline data and uses RLPD Ball et al. (2023) for policy training. *(iii)* **ExPLORe** (Li et al., 2023), which labels offline data using UCB rewards. We enhance the original implementation with reward ensembles. *(iv)* **JSRL-BC** (Uchendu et al., 2023), which collects online data using a mixture of the training policy and a behavior-cloned prior policy learned from offline data. As the compared baselines cannot handle multi-embodiment data like NCRL, we preprocess the offline data to only include task-relevant trajectories for them. Despite the baselines having access to better-structured data, NCRL still significantly outperforms all baselines across the tested tasks. See Sec. G for the details of baselines.

Fig. 3 (right) shows comparison results with baselines. Our method outperforms *all* compared baselines by a large margin. Compared to R3M, NCRL shows the importance of world model pre-training and reusing offline data during fine-tuning, versus representation learning alone. R3M fails to improve sample efficiency on most tasks, consistent with findings in Hansen et al. (2023).

UDS and ExPLORe reuse offline data by labeling it with zero rewards and UCB rewards, respectively, and concatenating it with online data for off-policy updates. UDS shows only slightly better performance on Walker Run compared to R3M and JSRL-BC, demonstrating the ineffectiveness of zero-reward labeling. ExPLORe performs better on 2/3 locomotion tasks and shows progress on challenging manipulation tasks, but NCRL still significantly outperforms it, demonstrating the superiority of leveraging a pre-trained world model and properly reusing offline data during fine-tuning.

NCRL also clearly outperforms JSRL-BC. JSRL-BC's performance heavily depends on the offline data distribution. While JSRL-BC can perform well when a good prior actor can be extracted from offline data, it struggles with non-expert trajectories, showing only marginal improvements over other baselines on the Cheetah Run Hard, Assembly, and Shelf Place tasks. In contrast, NCRL effectively leverages non-expert offline data. For example, on Quadruped Walk, NCRL benefits from exploratory offline data, enabling pixel-based control within just 100 trials.

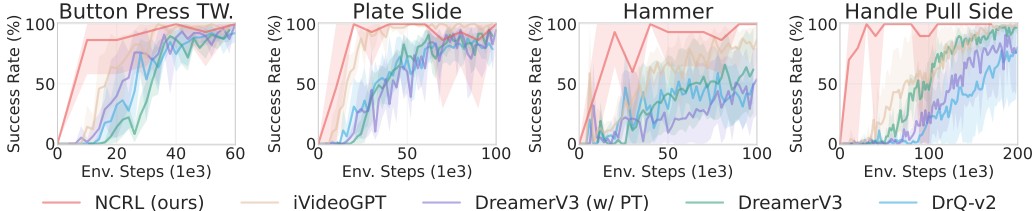

Figure 4: Comparison with other world model pre-training methods. NCRL outperforms state-of-the-art model-based methods without relying on techniques used in iVideoGPT, such as reward shaping and demonstration-based replay buffer initialization.

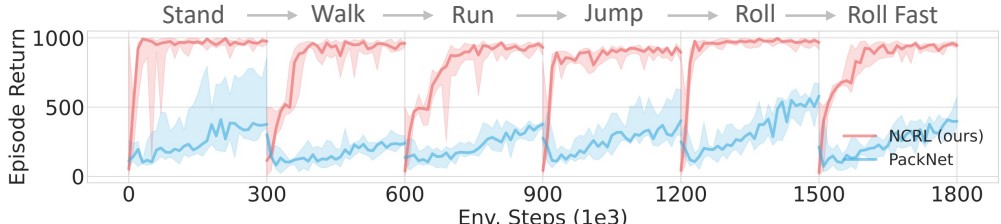

Figure 5: NCRL enables fast task adaptation. We train an RL agent to control an Ant robot from DMControl to complete a series of tasks incrementally. NCRL significantly outperforms the widely used baseline PackNet by properly leveraging non-curated offline data.

**Comparison with Training-from-Scratch Methods** We compare NCRL with two widely used training-from-scratch baselines: **DrQ-v2** and **DreamerV3**, representing model-free and model-based approaches, respectively. Fig. 3 (left) and Sec. I show comparison results on 22 locomotion and 50 robotic manipulation tasks with pixel inputs from DMControl and Meta-World benchmarks. With 150k online samples, NCRL achieves higher aggregate scores compared to DrQ-v2 and DreamerV3, matching their performance obtained with 3.3-6.7× more samples (500k for DMControl, 1M for Meta-World). Furthermore, NCRL achieves promising performance on hard exploration tasks where learning-from-scratch baselines fail, such as challenging Meta-World manipulation tasks and hard DMControl tasks.

**Comparison with Other Model-Based Methods** While most multi-task/multi-embodiment world models focus on visual prediction (Zheng et al., 2024; Zhu et al., 2024b) or imitation learning (Zhu et al., 2024a; Zhou et al., 2024), some works like Seo et al. (2022), Wu et al. (2024), and iVideoGPT (Wu et al., 2025) investigate world model pre-training with in-the-wild data for RL. These methods typically focus on designing novel or scalable model architectures to leverage the offline data, but lack mechanisms to better leverage offline data during RL fine-tuning. Furthermore, due to the cost of RL training, these methods are usually evaluated on limited task sets, making the effectiveness of the pre-trained world model unclear on diverse tasks.

Figure 4 compares our method with world model pre-training approaches. The baseline results are from the iVideoGPT paper to get the best reported results in the original paper. We further compare iVideoGPT in an aligned setting in Sec. A.2. Despite extensive pre-training on diverse manipulation data, iVideoGPT and pre-trained DreamerV3 show only marginal improvements over training-from-scratch baselines. In contrast, NCRL clearly accelerates RL training by properly leveraging non-curated offline data during both pre-training and fine-tuning. Notably, baselines in Fig. 4 use reward shaping and expert replay buffer pre-filling, while NCRL uses *none* of these tricks yet achieves superior performance. This highlights that *(i)* non-curated offline data contains useful information for RL fine-tuning, and *ii)* NCRL can effectively leverage such data. Furthermore, NCRL could potentially be combined with iVideoGPT to leverage even more diverse offline data in future work.

## 4.2 NCRL ENABLES FAST TASK ADAPTATION

We investigate NCRL's benefits for continual task adaptation, where an agent must incrementally solve a sequence of tasks. While similar to continual reinforcement learning (CRL) or life-long RL (Parisi et al., 2019; Khetarpal et al., 2022), we use a simplified setting with a limited task set.

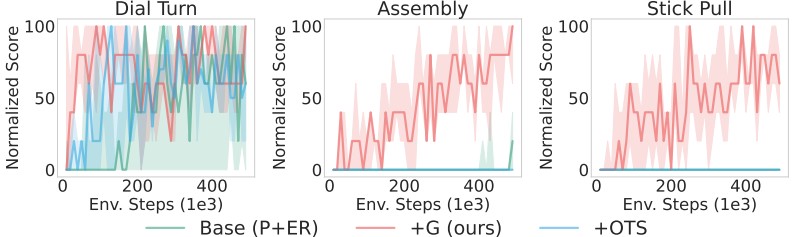

Figure 6: Ablation study on key components. "P" represents world model pre-training, "ER" means experience rehearsal, and "G" represents execution guidance. The combination of a pre-trained task-agnostic world model with retrieval-based experience rehearsal and execution guidance boosts RL performance across diverse tasks.

Figure 7: Comparison of execution guidance versus uncertainty-based reward labeling. NCRL demonstrates the effectiveness of using execution guidance over uncertainty-based reward labeling on challenging robotic manipulation tasks.

Note that CRL has a broad scope; assumptions and experiment setups vary among methods, making it difficult to set up a fair comparison with other methods. Rather than proposing a state-of-the-art CRL method, we aim to demonstrate that NCRL offers an effective approach to leverage previous data that also fits the CRL setting.

**Setup & Baselines**   We set our continual adaptation experiment based on the Quadruped robot from DMControl. Specifically, the agent sequentially learns stand, walk, run, jump, roll, and roll fast tasks with 300K environment steps per task. To have a fair comparison, i.e., having comparable model parameters and eliminating the potential effects from pre-training on other tasks, we pre-train a small world model only on the Quadruped domain. During training, the agent can access all previous experiences and model weights. We compare against a widely used baseline PackNet (Mallya & Lazebnik, 2018), which iteratively prunes actor parameters while preserving important weights to remember previous skills. For each new task, PackNet fine-tunes the actor model via iterative pruning while randomly reinitializing the critic model since rewards are not shared among tasks.

**Results**   Figure 5 shows NCRL significantly outperforms PackNet, enabling adaptation within 100 trials per task. With limited samples, PackNet achieves only 20–60% of NCRL's episodic returns. We attribute NCRL's superior performance to its ability to leverage the diverse offline data through both world model pre-training and fine-tuning with experience rehearsal and execution guidance.

## 4.3   ABLATIONS

**Role of Each Component**   We now analyze each component's contribution using the same set of tasks from Sec. 4.1. As shown in Fig. 6, world model pre-training shows promising results when the offline data consists of diverse trajectories, such as data collected by exploratory agents (Walker Run), while it fails to work well when the offline data distribution is relatively narrow as in the Meta-World tasks. We found that experience rehearsal and execution guidance stabilize training and improve performance on hard exploration tasks like Cheetah Run Hard and challenging manipulation tasks from Meta-World. This addresses *(i)* world model pre-training alone, failing to fully leverage rich state and action information from the non-curated offline data and *(ii)* distributional shift between offline and online data during fine-tuning hurts the learning. The proposed retrieval-based experience rehearsal and execution guidance help utilize offline data and accelerate exploration, which together enable NCRL to achieve strong performance on a wide range of tasks.

**Comparison with Uncertainty-Aware Reward Function** To leverage reward-free offline data, ExPLORe (Li et al., 2023) proposes to label offline data with uncertainty-based rewards. To demonstrate the effectiveness of NCRL, we compare it with uncertainty-based rewards. Specifically, instead of using execution guidance, we use Optimistic Thompson Sampling (OTS) (Hu et al., 2023b) to label the imagined trajectories via model rollout. As shown in Fig. 7, our method outperforms the variant using OTS on hard exploration tasks, Assembly and Stick Pull, by a large margin, showing the effectiveness of using execution guidance.

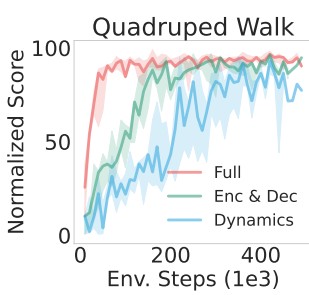

Figure 8: Impact of fine-tuning different world model components.

**Comparison of Fine-Tuning Different Components** We now investigate the role of different components in the world model during fine-tuning. We use the Quadruped Walk task as a representative task for the investigation. As shown in Fig. 8, the encoder, decoder, and latent dynamics play important roles during fine-tuning. Fine-tuning the full world model yields the best performance on the tested task. The full world model is fine-tuned by default in our experiments.

## 5 CONCLUSION

We propose NCRL, a simple yet efficient approach to leverage ample non-curated offline datasets consisting of reward-free, mixed-quality data collected across multiple embodiments. NCRL pre-trains a task-agnostic world model on the non-curated data and adapts to downstream tasks via RL. We show that naive fine-tuning of world models fails to accelerate RL training due to distributional shift and propose two techniques – experience rehearsal and execution guidance – to mitigate this issue. Equipped with these techniques, we demonstrate that world models pre-trained on non-curated data are able to boost RL's sample efficiency across a broader range of locomotion and robotic manipulation tasks. We compared NCRL against a wide set of baselines, including two widely used training-from-scratch methods, five methods that utilize offline data, and one continual learning method. Our NCRL consistently delivers strong performance over these baselines. Extensive ablation studies reveal the effectiveness of the proposed techniques. While promising, NCRL can be improved in multiple ways: extending to real-world applications, leveraging in-the-wild offline data, and exploring novel world model architectures.

### ETHICS STATEMENT

This paper contributes to the field of reinforcement learning (RL), with potential applications including robotics and autonomous machines. While our methods hold promise for advancing technology, they could also be applied in ways that raise ethical concerns, such as in autonomous machines exploring the world and making decisions on their own. However, the specific societal impacts of our work are broad and varied, and we believe a detailed discussion of potential negative uses is beyond the scope of this paper. We encourage a broader dialogue on the ethical use of RL technology and its regulation to prevent misuse.

### REPRODUCIBILITY STATEMENT

To facilitate reproducibility, we provide implementation descriptions in Sec. G, report computational requirements in Sec. F, present the complete algorithm in Sec. H, and specify key hyperparameters in Sec. J. Code and datasets are in https://github.com/zhaoyi11/ncrl.

### ACKNOWLEDGMENTS

We acknowledge CSC – IT Center for Science, Finland, for awarding this project access to the LUMI supercomputer, owned by the EuroHPC Joint Undertaking, hosted by CSC (Finland) and the LUMI consortium through CSC. We acknowledge the computational resources provided by the Aalto Science-IT project. We acknowledge funding from the Research Council of Finland (353138, 362407, 352788, 357301, 339730). Aidan Scannell and Wenshuai Zhao were supported by the Research Council of Finland, Flagship program Finnish Center for Artificial Intelligence (FCAI).

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

# Appendices

## A    MORE RESULTS

### A.1    COMPARISON WITH IMITATION LEARNING BASELINE

To demonstrate the mixed-quality property of the non-curated dataset, we compare NCRL with Diffusion Policy, a widely used imitation learning approach by modeling the agent with diffusion models. From Fig. 9, we can see that due to the dataset consisting of non-expert data, the diffusion policy fails to demonstrate satisfactory results, while NCRL can effectively utilize the offline data.

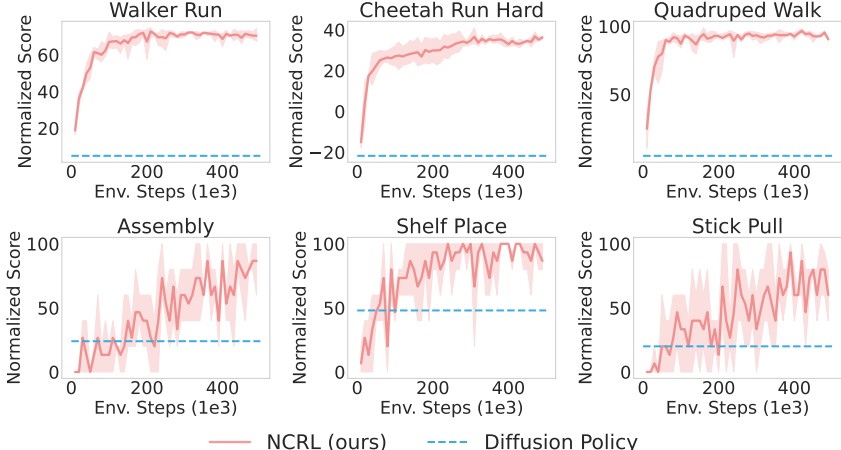

Figure 9: Comparison with Diffusion Policy. NCRL can effectively handle non-curated offline data while the imitation learning baseline fails.

### A.2    COMPARISON WITH iVIDEOGPT

**Comparison in an Aligned Setting**

In Fig. 4, we compare our method against the original iVideoGPT results (Wu et al., 2025). Their experimental setups differ than ours in several ways: *i)* iVideoGPT modifies the reward function to assign high rewards to successful episodes and *ii)* pre-fills the replay buffer with a few demonstrations to ease exploration. In addition, iVideoGPT is pre-trained on X-embodiment datasets (O'Neill et al., 2023), whereas our method uses data from the same domain as the downstream tasks.

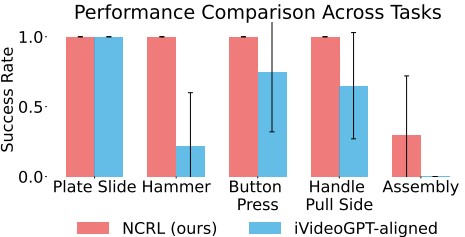

Figure 10: Comparison with aligned iVideoGPT.

To control for these differences, we run an additional set of experiments. Specifically, we fine-tune iVideoGPT on our dataset, initialize the policy with behavior cloning, and remove both reward shaping and demonstration pre-filling. We refer to this variant as iVideoGPT-align. We compare iVideoGPT-align with our NCRL after training with 200k environment steps Fig. 10, NCRL still outperforms iVideoGPT-align with a decent margin.

**Full Results of Comparison with iVideoGPT**    We compare with other model-based approaches on tasks used in iVideoGPT (Wu et al., 2025). We show that NCRL outperforms the baselines without using reward shaping and pre-filling the replay buffer with demonstrations. This highlights that although non-curated, the offline data can clearly boost RL training, and NCRL can effectively use the information in the data.

### A.3    EXPERIENCE RETRIEVAL PERFORMANCE

In Sec. 3.3, we adopt a simple criterion to retrieve task-relevant trajectories from the non-curated dataset. We evaluate retrieval performance in Table 2, reporting precision at the top-250 and top-

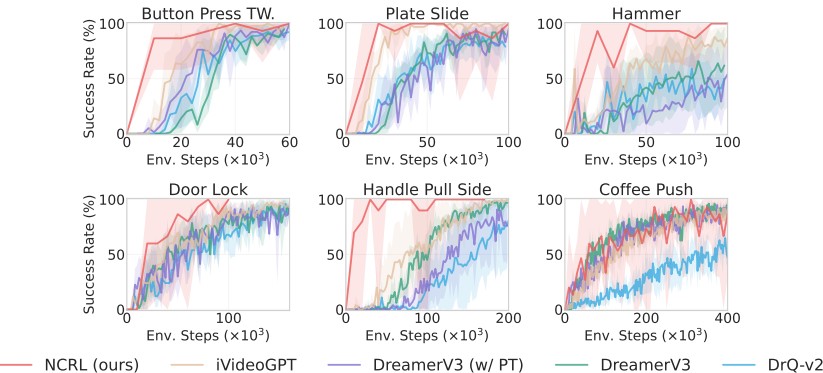

Figure 11: Comparison with model-based approaches for leveraging offline data.

500 retrieved trajectories. Our method achieves consistently high precision. For the Door Open task, some retrieved trajectories overlap with related tasks (Door Close, Door Lock, Door Unlock), but we find that RL training remains effective across all 72 evaluated tasks. A likely reason is that most RL training data is collected online, with policy and value functions updated from imaginary data generated by model rollouts, which mitigates the impact of occasional task-irrelevant trajectories. We expect future work to explore more advanced retrieval strategies for improved robustness.

Table 2: Precision results across tasks.

| Tasks | Quadruped Run | Assembly | Shelf Place | Door Open |
|---|---|---|---|---|
| Precision@250 | 100% | 100% | 100% | 84% |
| Precision@500 | 100% | 100% | 100% | 68% |

## A.4 MORE ABLATION STUDIES

**Hyperparameter Sensitivity** In execution guidance, we randomly sample both the starting timestep $t_{\text{start}}$ and duration $H$. Unlike JSRL (Uchendu et al., 2023), our approach eliminates expensive tuning for these hyperparameters and demonstrates robust performance. In this stage, we only introduce one hyperparameter to probabilistically decide whether to use $\pi_{\text{BC}}$ based on a linear annealing schedule. We show that in Fig. 12, our method is not sensitive to this annealing schedule, showing robustness in a wide range of possible schedules.

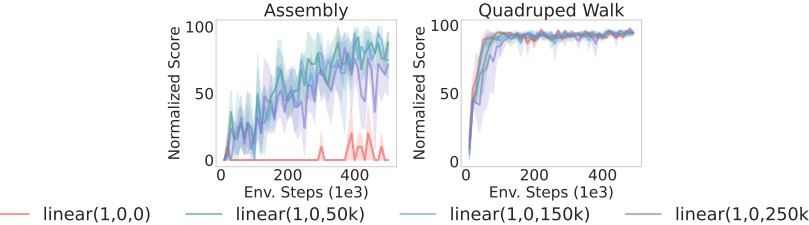

Figure 12: Our method is less sensitive to the choice of the execution guidance annealing schedule.

**Role of Each Component** We show inter-quartile mean (IQM) and optimality gap for the ablation study of the role of each proposed component in Fig. 13. Together with the retrieval-based experience rehearsal and execution guidance, a pre-trained task-agnostic world model boosts RL performance on a wide range of tasks.

**Impact of Retrieved Data** In Fig. 14, we evaluate the impact of retrieved data on the agent's performance to assess the robustness of NCRL with respect to the quality of the retrieved dataset. As shown in the experiments on three challenging MetaWorld manipulation tasks, we progressively replaced the retrieved task-relevant trajectories with 0%, 25%, 50%, 75%, and 100% trajectories that lie far from the target task in the latent space. We observed that our method remains robust even as the quality of the retrieved data degrades.

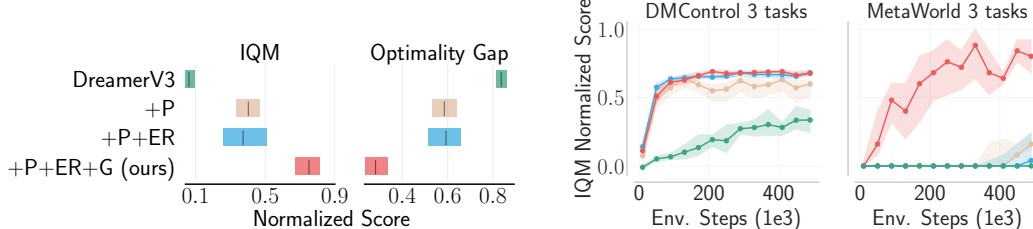

Figure 13: Ablation study on the role of each component. "P" represents world model pretraining, "ER" means experience rehearsal, and "G" represents execution guidance. Together with the proposed retrieval-based experience rehearsal and execution guidance, world model pre-training boosts RL performance on a wide range of tasks.

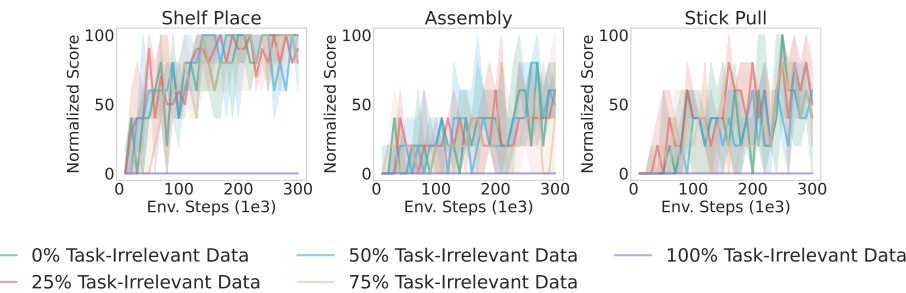

Figure 14: Comparison with injecting different ratios of task-irrelevant offline data. Our method remains robust even as the quality of the retrieved data degrades.

## A.5 MODEL SIZE OF DREAMERV3

In Sec. I, we compare NCRL with the DreamerV3 baseline under a commonly used but relatively small model-size configuration. Although DreamerV3 has shown performance gains on more challenging domains such as Craft and DMLab when using larger models, these benefits are less pronounced in the settings examined in this work (DMControl and MetaWorld). Indeed, DreamerV3 itself uses a relatively small model for DMControl tasks Hafner et al. (2023). To ensure a fair comparison, we additionally evaluated DreamerV3 using the same model size as NCRL. As shown in Fig. 15, increasing the model size improves DreamerV3's performance on Walker Run but degrades performance on Quadruped Walk. However, NCRL consistently outperforms the DreamerV3 baseline across different model sizes.

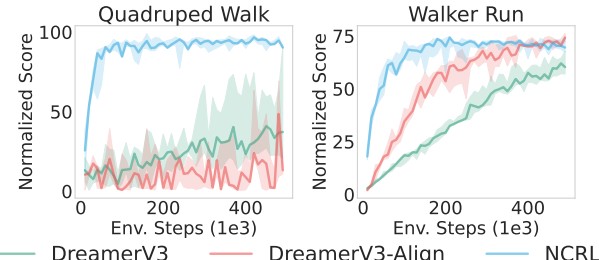

Figure 15: Comparison of DreamerV3 under different model size configurations. NCRL consistently outperforms both variants.

## A.6 PERFORMANCE ON CHALLENGING METAWORLD TASKS

In Sec. I, although NCRL solves most MetaWorld tasks with satisfactory performance, a few tasks still exhibit relatively low success rates with 150k environment steps. These tasks typically involve long horizons, small objects of interest, or strict success criteria. We have already shown three of these challenging tasks in Fig. 3, showing increased success rates with a larger training budget.

We now include additional experiments on other selected tasks using an increased training budget in Fig. 16. We found that, for most tasks, the success rate improves as the training budget increases.

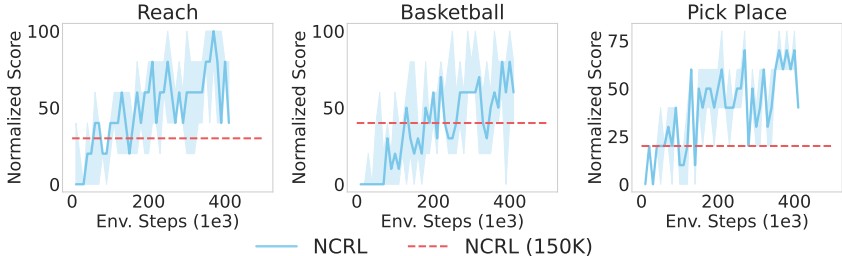

Figure 16: Improved success rate on MetaWorld tasks as the training budget increases.

# B THEORETICAL ANALYSIS

In this section, we give a theoretical analysis of the main conclusions in our paper.

## B.1 PROOF OF THE BENEFITS OF EXPERIENCE RETRIEVAL

**Proposition 1.** *Experience retrieval reduces distribution shift during online fine-tuning, compared to using the full offline dataset directly, in the sense that*

$$\mathbb{E}_{s\sim p_{retrieved},\ s_{on}\sim p_{on}}\left[||s - s_{on}||_2\right] < \mathbb{E}_{s\sim p_{off},\ s_{on}\sim p_{on}}\left[||s - s_{on}||_2\right]. \tag{6}$$

*Proof.* Let $p_{\text{off}}(s)$, $p_{\text{on}}(s)$, and $p_{\text{retrieved}}(s)$ denote the state distributions of the non-curated offline dataset $\mathcal{D}_{\text{off}}$, the online dataset $\mathcal{D}_{\text{on}}$, and the retrieved dataset $\mathcal{D}_{\text{retrieved}} \subset \mathcal{D}_{\text{off}}$, respectively. We simplify the notation as

$$\mathbb{E}_{s\sim p,\ s_{on}\sim p_{on}}\left[||s - s_{on}||_2\right] \quad \text{as} \quad \mathbb{E}_{s\sim p}\left[d(s, s_{on})\right].$$

Since $\mathcal{D}_{\text{retrieved}} \subset \mathcal{D}_{\text{off}}$, the distribution $p_{\text{off}}(s)$ can be expressed as a mixture distribution:

$$p_{\text{off}}(s) = \alpha \cdot p_{\text{retrieved}}(s) + (1 - \alpha) \cdot p_{\text{rest}}(s),$$

where $p_{\text{rest}}(s)$ is the distribution over the remaining offline data, and $\alpha = \frac{|\mathcal{D}_{\text{retrieved}}|}{|\mathcal{D}_{\text{off}}|}$ denotes the fraction of samples in the retrieved dataset.

The expected total variation for the mixture distribution decomposes as:

$$\mathbb{E}_{s\sim p_{\text{off}}}\left[d(s, s_{\text{on}})\right] = \alpha \cdot \mathbb{E}_{s\sim p_{\text{retrieved}}}\left[d(s, s_{\text{on}})\right] + (1 - \alpha) \cdot \mathbb{E}_{s\sim p_{\text{rest}}}\left[d(s, s_{\text{on}})\right]. \tag{7}$$

Assume that $\mathcal{D}_{\text{retrieved}}$ is constructed by selecting states such that $\|s_{\text{retrieved}} - s_{\text{on}}\| < \epsilon$, for some small $\epsilon > 0$. Consequently, states in $\mathcal{D}_{\text{rest}}$ satisfy $\|s_{\text{rest}} - s_{\text{on}}\| \geq \epsilon$. This construction implies the following bounds:

$$\mathbb{E}_{s\sim p_{\text{retrieved}}}\left[d(s, s_{\text{on}})\right] < \epsilon', \tag{8}$$

$$\mathbb{E}_{s\sim p_{\text{rest}}}\left[d(s, s_{\text{on}})\right] \geq \epsilon', \tag{9}$$

for some $\epsilon' > 0$. Therefore, it follows that

$$\mathbb{E}_{s\sim p_{\text{rest}}}\left[d(s, s_{\text{on}})\right] > \mathbb{E}_{s\sim p_{\text{retrieved}}}\left[d(s, s_{\text{on}})\right].$$

Substituting into Equation equation 7 yields:

$$\begin{aligned}
\mathbb{E}_{s\sim p_{\text{off}}}\left[d(s, s_{\text{on}})\right] &= \alpha \cdot \mathbb{E}_{s\sim p_{\text{retrieved}}}\left[d(s, s_{\text{on}})\right] + (1 - \alpha) \cdot \mathbb{E}_{s\sim p_{\text{rest}}}\left[d(s, s_{\text{on}})\right] \\
&> \alpha \cdot \mathbb{E}_{s\sim p_{\text{retrieved}}}\left[d(s, s_{\text{on}})\right] + (1 - \alpha) \cdot \mathbb{E}_{s\sim p_{\text{retrieved}}}\left[d(s, s_{\text{on}})\right] \\
&= \mathbb{E}_{s\sim p_{\text{retrieved}}}\left[d(s, s_{\text{on}})\right].
\end{aligned}$$

Thus, the expected total variation between the retrieved data and online data is strictly smaller than that between the full offline data and online data. $\square$

**Explanation.** Experience retrieval helps prevent catastrophic forgetting during online fine-tuning.

**Definition B.1** (Catastrophic Forgetting due to Data Distribution Shift). Catastrophic forgetting occurs when a neural network, after training on a new data distribution, experiences a significant performance drop on previously learned tasks due to the overwriting of representations from earlier distributions, caused by biased parameter updates towards the new distribution.

*Proof.* Following the previous notations, let $\mathcal{D}_{\text{on}}$ and $\mathcal{D}_{\text{retrieved}}$ denote the online dataset and the retrieved offline dataset, respectively. The objective in Eq. (1) can be written as:

$$\mathcal{L}_{\text{mixed}}(\theta) = \mathcal{L}_{\text{on}}(\theta) + \lambda \cdot \mathcal{L}_{\text{retrieved}}(\theta)$$

$$= \mathbb{E}_{p_\theta, q_\theta, (o,a) \sim \mathcal{D}_{\text{on}}} \left[ \sum_{t=1}^{T} - \ln p_\theta(o_t \mid z_t, h_t) + \beta \cdot \text{KL} \left( q_\theta(z_t \mid h_t, o_t) \parallel p_\theta(z_t \mid h_t) \right) \right]$$

$$+ \lambda \cdot \mathbb{E}_{p_\theta, q_\theta, (o,a) \sim \mathcal{D}_{\text{retrieved}}} \left[ \sum_{t=1}^{T} - \ln p_\theta(o_t \mid z_t, h_t) + \beta \cdot \text{KL} \left( q_\theta(z_t \mid h_t, o_t) \parallel p_\theta(z_t \mid h_t) \right) \right].$$

Assuming the $\lambda$ is a monotonic function of $\alpha = \frac{|\mathcal{D}_{\text{retrieved}}|}{|\mathcal{D}_{\text{off}}|}$ and $\lambda > 0$, since $\mathcal{D}_{\text{retrieved}} \subset \mathcal{D}_{\text{off}}$, the term $\mathcal{L}_{\text{retrieved}}(\theta)$ acts as a regularizer during online updates, constraining parameter changes on $\mathcal{D}_{\text{on}}$ in a way that preserves performance on the retrieved offline distribution $p_{\text{retrieved}}$. This mitigates the risk of catastrophic forgetting by anchoring the model to previously seen data. $\square$

## B.2 PROOF OF IMPROVED PERFORMANCE WITH EXECUTION GUIDANCE

**Proposition 2** (Performance Improvement via Execution Guidance). *Let $\pi^e$ denote an exploration policy and $\pi^g$ a guide policy obtained via imitation learning. Let $\varepsilon = \max_s |\mathbb{E}_{a \sim \pi^g(\cdot|s)}[A_{\pi^e}(s, a)]|$.*

*Let $\tilde{\pi}$ be a mixed policy (execution guidance) derived from $\pi^e$ and $\pi^g$, defined as:*

$$\tilde{\pi}(a|s) = \alpha \pi^g(a|s) + (1 - \alpha)\pi^e(a|s), \ \alpha \in [0, 1]. \tag{10}$$

*Then, the performance of the mixed policy $\tilde{\pi}$ exceeds that of the exploration policy $\pi^e$ by at least:*

$$\eta(\tilde{\pi}) - \eta(\pi^e) \geq \frac{\alpha}{1 - \gamma} E_{s \sim d_\pi} \left[ \sum_a \pi^g(\cdot|s) A_\pi(s_t, a) \right] \tag{11}$$

$$- 2\alpha\varepsilon \left( \frac{1}{1 - \gamma} - \frac{1}{1 - \gamma(1 - \alpha)} \right).$$

*where $\gamma \in [0, 1)$ is the discount factor.*

*Proof.* The proof follows directly from Theorem 4.1 in Kakade & Langford (2002): we just need to replace $\pi'$ in Kakade & Langford (2002) with $\pi^g$ and $\pi$ in Kakade & Langford (2002) with $\pi^e$. According to (Kakade & Langford, 2002) an example can be provided where this bound is tight.

This establishes that the performance improvement of the mixed policy $\tilde{\pi}$ over the exploration policy $\pi^e$ is positive when the expected policy improvement of the guidance policy over the execution policy $E_{s \sim d_{\pi^e}} \left[ \sum_a \pi^g(\cdot|s) A_{\pi^e}(s_t, a) \right]$ is larger than the term $-2(1 - \gamma)\varepsilon(\frac{1}{1-\gamma} - \frac{1}{1-\gamma(1-\alpha)})$ which results from the distribution shift due to using the guidance policy instead of only the execution policy.

Moreover, according to Corallary 4.2 in Kakade & Langford (2002), when $E_{s \sim d_{\pi^e}} \left[ \sum_a \pi^g(\cdot|s) A_{\pi^e}(s_t, a) \right] \geq 0$ and when the maximal immediate reward is positive, we can always choose $\alpha$ such that the performance improvement is positive (see Kakade & Langford (2002) for details). $\square$

## C   MORE RELATED WORK

In this section, we give a more detailed related work review.

**RL with task-specific offline datasets**   Leveraging offline data is a promising direction to improve sample efficiency in RL. One representative approach is offline RL, which trains agents using offline data without environment interaction. These methods typically constrain the distance between learned and behavior policies in different ways (Kumar et al., 2020; Fujimoto & Gu, 2021; Kumar et al., 2019; Wu et al., 2019; Kostrikov et al., 2021; 2022; Uchendu et al., 2023). However, policy performance is highly dependent on dataset quality (Yarats et al., 2022). To enable continued improvement, offline-to-online RL methods (Lee et al., 2022; Zhao et al., 2022; Yu & Zhang, 2023; Nair et al., 2020; Rafailov et al., 2023) were developed, which fine-tune policies trained with offline RL by interacting with environments. MOTO (Rafailov et al., 2023) proposes a model-based offline-to-online RL method with reward-labeled data, and requires model-based value expansion, policy regularization, and controlling epistemic uncertainty, while our method leverages reward-free and multi-embodiment data and requires none of the techniques proposed by MOTO.

Typical offline-to-online RL face training instability challenges (Lee et al., 2022; Lu et al., 2023). To mitigate this issue, RLPD (Ball et al., 2023) is proposed and demonstrates strong performance by simply concatenating offline and online data, but requires reward-labeled task-specific offline data and does not address multi-embodiment scenarios. ExPLORe (Li et al., 2023) labels reward-free offline data using approximated upper confidence bounds (UCB) to solve hard exploration tasks, but relies on near-expert data for the target tasks, while we consider a more general setting with non-curated data.

**RL with multi-task offline datasets**   Recent work has explored multi-task offline RL (Kumar et al., 2023; Hansen et al., 2024; Julian et al., 2020; Kalashnikov et al., 2021; Yu et al., 2021), but requires known rewards. PWM (Georgiev et al., 2024) and TDMPC-v2 (Hansen et al., 2024) train world models for multi-task RL but are limited to state-based inputs and reward-labeled data. To handle unknown rewards, approaches like human labeling (Cabi et al., 2020; Singh et al., 2019), inverse RL (Ng et al., 2000; Abbeel & Ng, 2004), or generative adversarial imitation learning (Ho & Ermon, 2016) can be used, though these require human labor or expert demonstrations. Yu et al. (2022) assigns zero rewards to unlabeled data, which introduces additional bias. Apart from these, there is a line of work that focuses on representation learning from in-the-wild data (Schwarzer et al., 2021; Parisi et al., 2022; Yang & Nachum, 2021; Yuan et al., 2022; Stooke et al., 2021; Shah & Kumar, 2021; Wang et al., 2022; Sun et al., 2023; Ze et al., 2023; Ghosh et al., 2023) but fails to utilize rich information in the dataset, such as dynamics.

Recent studies (Seo et al., 2022; Wu et al., 2025; 2023) explore world model pre-training with action-free data, focusing on world model architecture design to utilize the action-free data. However, we demonstrate that naive fine-tuning of pre-trained world models fails on challenging tasks, while our method, incorporating experience rehearsal and execution guidance, significantly improves RL performance across 72 tasks.

**Unsupervised RL**   In unsupervised RL, an agent explores the environment based on intrinsic motivations, and the models' parameters are initialized during this self-motivated exploration stage, aiming for fast downstream task learning (Rajeswar et al., 2023; Pathak et al., 2017; Burda et al., 2019; Eysenbach et al., 2019; Pathak et al., 2019; Liu & Abbeel, 2021a; Sekar et al., 2020; Liu & Abbeel, 2021b; Yarats et al., 2021; Laskin et al., 2021; Mazzaglia et al., 2023; Xu et al., 2022). Our problem setting differs from unsupervised RL in several ways: i) Unsupervised RL interacts with the environment actively while we leverage *static* offline datasets, ii) unsupervised RL gives a specific focus on designing different intrinsic rewards, while our setting focuses on improving sample efficiency by leveraging unlabeled datasets.

**Generalist Agents**   RL methods usually perform well on a single task (Vinyals et al., 2019; Andrychowicz et al., 2020), however, this contrasts with humans who can perform multiple tasks well. Recent works have proposed generalist agents that master a diverse set of tasks with a single agent (Reed et al., 2022; Brohan et al., 2023b; Team et al., 2024; Zhao et al., 2024). These methods typically resort to scalable models and large datasets and are trained via imitation learning (Brohan et al., 2023b;a; O'Neill et al., 2023; Khazatsky et al., 2024). In contrast, we train a task-agnostic world model and use it to boost RL performance for multiple tasks and embodiments.

**World models** World models learn to predict future observations or states based on historical information. World models have been widely investigated in online model-based RL (Hafner et al., 2020; Ha & Schmidhuber, 2018; Micheli et al., 2023; Alonso et al., 2024; Scannell et al., 2025). Recently, the community has started investigating scaling world models (Ha & Schmidhuber, 2018), for example, Hu et al. (2023a); Pearce et al. (2024); Wu et al. (2025); Agarwal et al. (2025); Mereu et al. (2025) train world models with Diffusion Models or Transformers. However, these models are usually trained on demonstration data. In contrast, we explore the offline-to-online RL setting – closely fitting the pre-train and then fine-tune paradigm – and we focus on leveraging reward-free and multi-embodiment data to increase the amount of available data for pre-training. We further identify the distributional shift issue when fine-tuning the pre-trained world model and mitigate the issue by proposing experience rehearsal and execution guidance.

## D  LIMITATIONS

Although demonstrating strong performance on a diverse set of tasks, our method has the following limitations. *1)* The world model architecture used in our paper is the recurrent state space model. This model is built upon RNN, which can be limited for scaling. This can be mitigated by using a Transformer and a diffusion-based world model. However, we note that the main conclusion of this paper should still be valid. *2)* We do not thoroughly discuss the generalization ability of the pre-trained world model. With DMControl tasks, our method shows a promising trend in generalizing to unseen tasks. However, generalization to new embodiments or novel configurations is still challenging, which requires even diverse training data. *3)* The non-curated offline data used in our paper, although lifting several key assumptions in previous offline-to-online RL, is still in-domain data, i.e., our current method is not able to leverage the vast in-the-wild data. A promising direction is to combine in-the-wild data for pre-training as in (Wu et al., 2025) and the domain-specific "in-house" data (as used in our paper) for post-training. *4)* We only conduct experiments in the simulator. Considering the sample efficiency of our proposed method, it could be promising to conduct experiments on real-world applications.

## E  DISCLOSURE OF LLMS USAGE

Large Language Models (LLMs) were used to assist word choice, improving grammar as well as proof checking in Sec. B. LLMs were also used in compressing the Related Work section due to page limits. The Related Work section was initially written by authors without using LLMs and the compressed text was subsequently revised by the authors. The main draft was written by authors without using LLMs. The ideas were formalized independently of LLMs assistance.

## F  COMPUTE RESOURCES

We conduct all experiments on clusters equipped with AMD MI250X GPUs, 64-core AMD EPYC "Trento" CPUs, and 64 GBs DDR4 memory. For pre-training, it takes $\sim 48$ GPU hours for 150k steps. For fine-tuning, it tasks $\sim 8$ GPU hours per run for 150K environment steps. Note that due to AMD GPUs not supporting hardware rendering, the training time should be longer than using Nvidia GPUs. To reproduce the NCRL's results in Fig. 3, it roughly takes 8 h * 72 tasks * 3 seeds = 1728 GPU hours.

## G  IMPLEMENTATION DETAILS

### G.1  BEHAVIOR CLONING

The Behavior Cloning methods used in both the execution guidance of NCRL and JSRL-BC are the same. We use a four-layer convolutional neural network (LeCun et al., 1995) with kernel depth [32, 64, 128, 256] following a three-layer MLPs with LayerNorm (Ba et al., 2016) after all linear layers.

We list the adopted encoder and actor architectures for reference.

```
1  class Encoder(nn.Module):
2    def __init__(self, obs_shape):
3      super().__init__()
4      assert obs_shape == (9, 64, 64), f'obs_shape is {(obs_shape)}, but
       expect (9, 64, 64)' # inputs shape
5
6      self.repr_dim = (32 * 8) * 2 * 2
7      _input_channel = 9
8
9      self.convnet = nn.Sequential(
10       nn.Conv2d(_input_channel, 32, 4, stride=2), # [B, 32, 31, 31]
11         nn.ELU(),
12         nn.Conv2d(32, 32*2, 4, stride=2), #[B, 64, 14, 14]
13         nn.ELU(),
14         nn.Conv2d(32*2, 32*4, 4, stride=2), #[B, 128, 6, 6]
15         nn.ELU(),
16         nn.Conv2d(32*4, 32*8, 4, stride=2), #[B, 256, 2, 2]
17         nn.ELU())
18       self.apply(utils.weight_init)
19
20    def forward(self, obs):
21      B, C, H, W = obs.shape
22
23      obs = obs / 255.0 - 0.5
24      h = self.convnet(obs)
25      # reshape to [B, -1]
26      h = h.view(B, -1)
27      return h
```

```
1  class Actor(nn.Module):
2    def __init__(self, repr_dim, action_shape, feature_dim=50, hidden_dim
       =1024):
3      super().__init__()
4      self.trunk = nn.Sequential(nn.Linear(repr_dim, feature_dim),
5                                 nn.LayerNorm(feature_dim), nn.Tanh())
6
7      self.policy = nn.Sequential(nn.Linear(feature_dim, hidden_dim),
8                                  nn.LayerNorm(hidden_dim), nn.ELU(),
9                                  nn.Linear(hidden_dim, hidden_dim),
10                                 nn.LayerNorm(hidden_dim), nn.ELU(),
11                                 nn.Linear(hidden_dim, action_shape[0]))
12
13     self.apply(utils.weight_init)
14
15   def forward(self, obs, std):
16     h = self.trunk(obs)
17     return self.policy(h)
```

### G.2   JSRL+BC

Jump-start RL (Uchendu et al., 2023) is proposed as an offline-to-online RL method. It includes two policies, a prior policy $\pi_{\theta_1}(a|s)$ and a behavior policy $\pi_{\theta_2}(a|s)$, where the prior policy is trained via offline RL methods and the behavior policy is updated during the online learning stage. However, offline RL requires the offline dataset to include rewards for the target task. To extract behavior policy from the offline dataset, we use the BC agent described above as the prior policy. During online training, in each episode, we randomly sample the rollout horizon $h$ of the prior policy from a pre-defined array `np.arange(0, 101, 10)`. We then execute the prior policy for $h$ steps and switch to the behavior policy until the end of an episode.

### G.3 ExPLORe

For the ExPLORe baseline, we follow the original training code[2]. We sweep over several design choices: i) kernel size of the linear layer used in the RND and reward models: [256 (default), 512]; ii) initial temperature value: [0.1 (default), 1.0]; iii) whether to use LayerNorm Layer (no by default); iv) learning rate: [1e-4, 3e-4 (default)]. However, we fail to obtain satisfactory performance. There are several potential reasons: i) the parameters used in the ExPLORe paper are tuned specifically to their setting, where manipulation tasks and near-expert trajectories are used; ii) the coefficient term of the RND value needs to be tuned carefully for different tasks and the reward should also be properly normalized.

To achieve reasonable performance and eliminate the performance gap caused by implementation-level details, we make the following modifications: i) we replace the RND module with ensembles to calculate uncertainty; ii) the reward function shares the latent space with the actor and critic.

---

[2]Source code of ExPLORe https://github.com/facebookresearch/ExPLORe

# H  ALGORITHM

The full algorithm is described in Alg. 1.

---

**Algorithm 1** Efficient RL by Guiding World Models with Non-Curated Offline Data

---

**Require:** Non-curated offline data $\mathcal{D}_{\text{off}}$, Online data $\mathcal{D}_{\text{on}} \leftarrow \emptyset$, Retrieval data $\mathcal{D}_{\text{retrieval}} \leftarrow \emptyset$
  World model $f_\theta, q_\theta, p_\theta, d_\theta$
  Policy $\pi_{\phi_{\text{RL}}}, \pi_{\phi_{\text{BC}}}$, Value function $v_\phi$ and Reward $r_\xi$.

*// Task-Agnostic World Model Pre-Training*
**for** num. pre-train steps **do**
  Randomly sample mini-batch $\mathcal{B}_{\text{off}} : \{o_t, a_t, o_{t+1}\}_{t=0}^T$ from $\mathcal{D}_{\text{off}}$.
  Update world model $f_\theta, q_\theta, p_\theta, d_\theta$ by minimizing Eq. (1) on sampled batch $\mathcal{B}$.
**end for**

*// Task-Specific Training*
*// Experience Retrieval*
Collect one initial observation $o_{\text{on}}^0$ from the environment.
Compute the visual similarity between $o_{\text{on}}$ and initial observations of trajectories $o_{\text{off}}$ in $\mathcal{D}_{\text{off}}$ using Eq. (5).
Select R trajectories according to Eq. (5) and fill $\mathcal{D}_{\text{retrieval}}$.

*// Behavior Cloning Policy Training*
**for** num. bc updates **do**
  Randomly sample mini-batch $\mathcal{B}_{\text{retrieval}} : \{o_t, a_t\}_{t=0}^N$ from $\mathcal{D}_{\text{retrieval}}$.
  Update $\pi_{\phi_{\text{BC}}}$ by minimizing $-\frac{1}{N}\sum_{t=0}^N \log \pi_{\phi_{\text{BC}}}(a_t|o_t)$.
**end for**

*// Task-Specific RL Fine-Tuning*
**for** num. episodes **do**
  *// Collect Data*
  Decide whether to use $\pi_{\phi_{\text{BC}}}$ according to the predefined schedule.
  **if** Select $\pi_{\phi_{\text{BC}}}$ **then**
    Randomly select the starting time step $k$ and the rollout horizon $H$.
  **end if**
  $t \leftarrow 0$
  **while** $t \leq$ episode length **do**
    $a_t = \pi_{\phi_{BC}}(a_t|o_t)$ if Use $\pi_{\phi_{\text{BC}}}$ and $k \leq t \leq H$ else $a_t = \pi_{\phi_{RL}}(a_t|o_t)$.
    Interact with the environment using $a_t$. Store $\{o_t, a_t, r_t, o_{t+1}\}$ to $\mathcal{D}_{\text{on}}$.
    $t \leftarrow t + 1$
  **end while**

  *// Update Models*
  **for** num. grad steps **do**
    Randomly sample mini-batch $\mathcal{B}_{\text{on}} : \{o_t, a_t, r_t, o_{t+1}\}_{t=0}^T$ from $\mathcal{D}_{on}$ and $\mathcal{B}_{\text{retrieval}} : \{o_t, a_t, r_t, o_{t+1}\}_{t=0}^T$ from $\mathcal{D}_{\text{retrieval}}$.
    Update world model $f_\theta, q_\theta, p_\theta, d_\theta$ by minimizing Eq. (1) on sampled batch $\{\mathcal{B}_{\text{on}}, \mathcal{B}_{\text{retrieval}}\}$.
    Update $r_\xi$ by minimizing $-\frac{1}{N}\sum_{i=0}^N \log p_\xi(r_t|s_t)$ on $\mathcal{B}_{\text{on}}$.    ◁ $s_t = [h_t, z_t]$
    *// Update policy and value function*
    Generate imaginary trajectories $\tilde{\tau} = \{s_t, a_t, s_{t+1}\}_{t=0}^T$ by rolling out $h_\theta, p_\theta$ with $\pi_{\phi_{\text{RL}}}$.
    Update policy $\pi_{\phi_{\text{RL}}}$ and value function $v_\phi$ with Eq. (4).
  **end for**
**end for**

---

## I FULL RESULTS

In Table 3 and Table 4, we list the success rate of 50 Meta-World benchmark tasks with pixel inputs. In Table 5, we list the episodic return of DMControl of 22 tasks. We compare NCRL at 150k samples with two widely used baselines DreamerV3 and DrQ-v2 at both 150k samples and 1M samples. We report the results over 5 random seeds for NCRL and 3 random seeds for DreamerV3 and DrQ-v2. The best results are marked with a bold font at 150k samples and the highest overall scores are marked with underline. The detailed result curves of both Meta-World and DMControl are shown in Fig. 17, Fig. 18, and Fig. 19.

### I.1 META-WORLD BENCHMARK

Table 3: Success rate of Meta-World benchmark with pixel inputs.

| Tasks | DreamerV3 @ 1M | DrQ-v2 @ 1M | DreamerV3 @ 150k | DrQ-v2 @ 150k | NCRL @ 150k |
|---|---|---|---|---|---|
| Assembly | 0.0 | 0.0 | 0.0 | 0.0 | **0.44** |
| Basketball | 0.0 | 0.97 | 0.0 | 0.0 | **0.36** |
| Bin Picking | 0.0 | 0.93 | 0.0 | 0.33 | **0.84** |
| Box Close | 0.13 | 0.9 | 0.0 | 0.0 | **0.88** |
| Button Press | 1.0 | 0.7 | 0.47 | 0.13 | **0.76** |
| Button Press Topdown | 1.0 | 1.0 | 0.33 | 0.17 | **1.0** |
| Button Press Topdown Wall | 1.0 | 1.0 | 0.73 | 0.63 | **1.0** |
| Button Press Wall | 1.0 | 1.0 | 0.93 | 0.77 | **1.0** |
| Coffee Button | 1.0 | 1.0 | 1.0 | 1.0 | 1.0 |
| Coffee Pull | 0.6 | 0.8 | 0.0 | **0.6** | 0.56 |
| Coffee Push | 0.67 | 0.77 | 0.13 | 0.2 | **0.72** |
| Dial Turn | 0.67 | 0.43 | 0.13 | 0.17 | **0.65** |
| Disassemble | 0.0 | 0.0 | 0.0 | 0.0 | 0.0 |
| Door Close | - | - | - | - | 1.0 |
| Door Lock | 1.0 | 0.93 | 0.6 | **0.97** | 0.96 |
| Door Open | 1.0 | 0.97 | 0.0 | 0.0 | **0.92** |
| Door Unlock | 1.0 | 1.0 | **1.0** | 0.63 | 0.92 |
| Drawer Close | 0.93 | 1.0 | 0.93 | **1.0** | 0.92 |
| Drawer Open | 0.67 | 0.33 | 0.13 | 0.33 | **1.0** |
| Faucet Open | 1.0 | 1.0 | 0.47 | 0.33 | **1.0** |
| Faucet Close | 0.87 | 1.0 | **1.0** | **1.0** | 0.92 |
| Hammer | 1.0 | 1.0 | 0.07 | 0.4 | **1.0** |
| Hand Insert | 0.07 | 0.57 | 0.0 | 0.1 | **0.44** |
| Handle Press Side | 1.0 | 1.0 | 1.0 | 1.0 | **1.0** |
| Handle Press | 1.0 | 1.0 | 0.93 | 0.97 | **1.0** |
| Handle Pull Side | 0.67 | 1.0 | 0.67 | 0.6 | **1.0** |
| Handle Pull | 0.67 | 0.6 | 0.33 | 0.6 | **0.85** |
| Lever Pull | 0.73 | 0.83 | 0.0 | 0.33 | **0.72** |
| More results see Table 4 | | | | | |

Table 4: Success rate of Meta-World benchmark with pixel inputs (Cont.).

| Tasks | DreamerV3 @ 1M | DrQ-v2 @ 1M | DreamerV3 @ 150k | DrQ-v2 @ 150k | NCRL (ours) @ 150k |
|---|---|---|---|---|---|
| Peg Insert Side | 1.0 | 1.0 | 0.0 | 0.27 | **1.0** |
| Peg Unplug Side | 0.93 | 0.9 | **0.53** | 0.5 | 0.48 |
| Pick Out of Hole | 0.0 | 0.27 | 0.0 | 0.0 | **0.25** |
| Pick Place Wall | 0.2 | 0.17 | 0.0 | 0.0 | **0.64** |
| Pick Place | 0.67 | 0.67 | 0.0 | 0.0 | **0.20** |
| Plate Slide Back Side | 1.0 | 1.0 | 0.93 | 1.0 | **1.0** |
| Plate Slide Back | 1.0 | 1.0 | 0.8 | 0.97 | **1.0** |
| Plate Slide Side | 1.0 | 0.9 | **0.73** | 0.5 | 0.52 |
| Plate Slide | 1.0 | 1.0 | 0.93 | **1.0** | 0.95 |
| Push Back | 0.33 | 0.33 | 0.0 | 0.0 | **0.32** |
| Push Wall | 0.33 | 0.57 | 0.0 | 0.0 | **0.84** |
| Push | 0.26 | 0.93 | 0.0 | 0.13 | **0.72** |
| Reach | 0.87 | 0.73 | **0.67** | 0.43 | 0.40 |
| Reach Wall | 1.0 | 0.87 | 0.53 | 0.7 | **0.80** |
| Shelf Place | 0.4 | 0.43 | 0.0 | 0.0 | **0.80** |
| Soccer | 0.6 | 0.3 | 0.13 | 0.13 | **0.16** |
| Stick Push | 0.0 | 0.07 | 0.0 | 0.0 | **0.64** |
| Stick Pull | 0.0 | 0.33 | 0.0 | 0.0 | **0.52** |
| Sweep Into | 0.87 | 1.0 | 0.0 | **0.87** | 0.72 |
| Sweep | 0.0 | 0.73 | 0.0 | 0.3 | **0.64** |
| Window Close | 1.0 | 1.0 | 0.93 | **1.0** | **1.0** |
| Window Open | 1.0 | 0.97 | 0.6 | **1.0** | 0.96 |
| **Mean** | 0.656 | 0.753 | 0.360 | 0.430 | **0.748** |
| **Medium** | 0.870 | 0.900 | 0.130 | 0.330 | **0.840** |

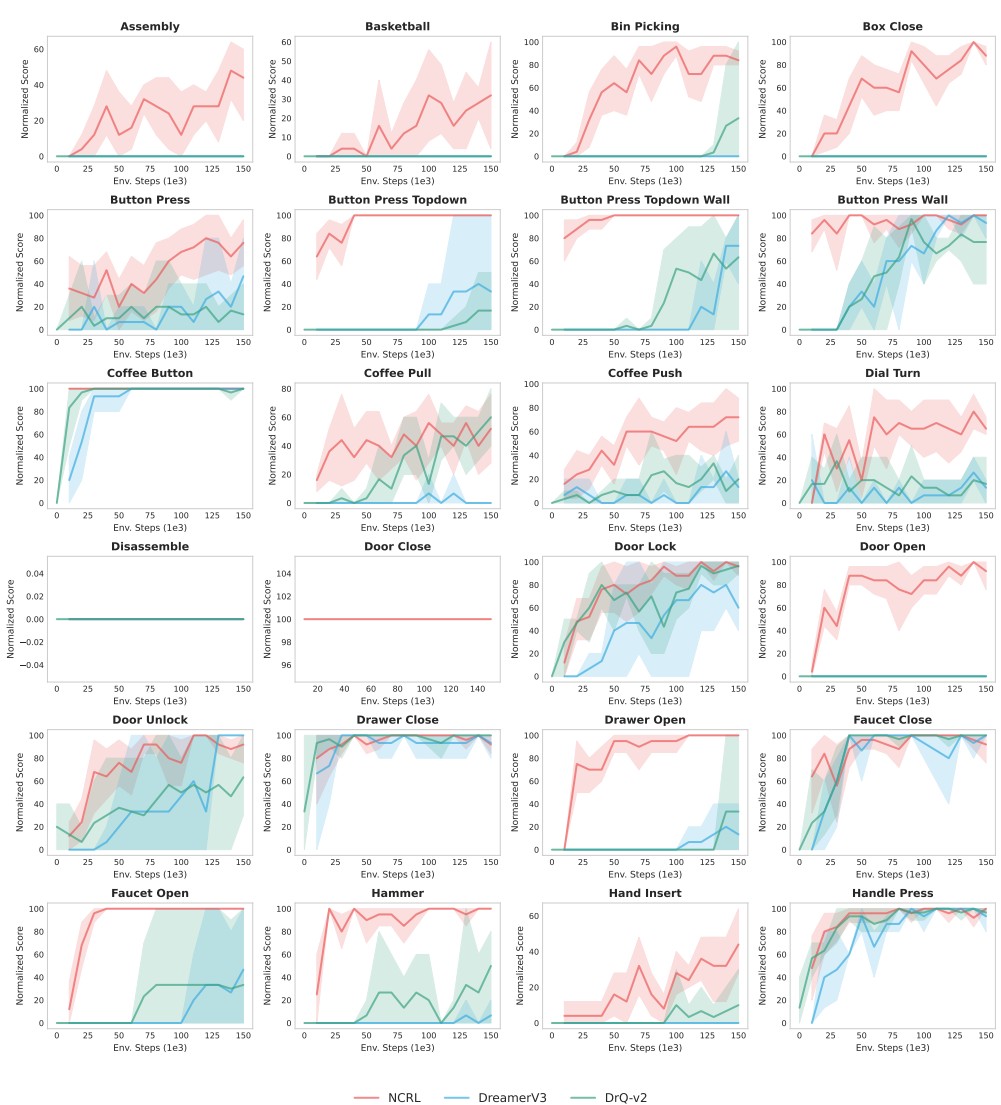

Figure 17: Meta-World results. We report 5 seeds for NCRL and 3 seeds for DrQ-v2 and DreamerV3.

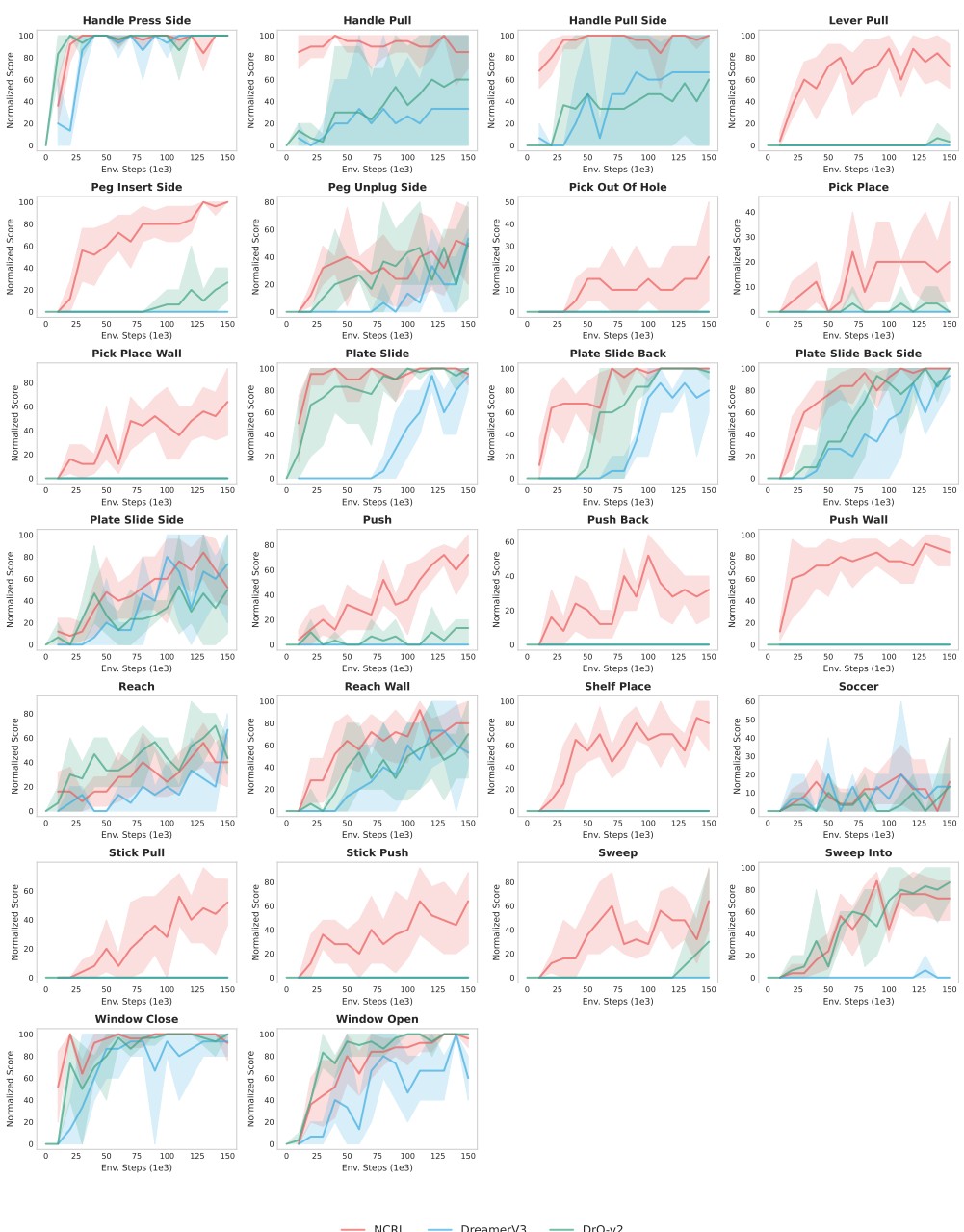

Figure 18: Meta-World results (Cont.). We report 5 seeds for NCRL and 3 seeds for DrQ-v2 and DreamerV3.

## I.2  DMCONTROL BENCHMARK

Table 5: Episodic return of DMControl benchmark with pixel inputs.

| Tasks | DreamerV3 @ 500k | DrQ-v2 @ 500k | DreamerV3 @ 150k | DrQ-v2 @ 150k | NCRL(ours) @ 150k |
|---|---|---|---|---|---|
| CartPole Balance | 994.3 | 992.3 | 955.8 | 983.3 | **995.0** |
| Acrobot Swingup | 222.1 | 30.3 | **85.2** | 20.8 | 84.6 |
| Acrobot Swingup Sparse | 2.5 | 1.17 | 1.7 | 1.5 | **12.2** |
| Acrobot Swingup Hard | -0.2 | 0.3 | **2.0** | 0.4 | -17.1 |
| Walker Stand | 965.7 | 947.6 | 946.2 | 742.9 | **974.1** |
| Walker Walk | 949.2 | 797.8 | 808.9 | 280.1 | **960.5** |
| Walker Run | 616.6 | 299.3 | 224.4 | 143.0 | **707.7** |
| Walker Backflip | 293.6 | 96.7 | 128.2 | 91.7 | **266.1** |
| Walker Walk Backward | 942.9 | 744.3 | 625.9 | 470.9 | **887.6** |
| Walker Walk Hard | -2.1 | -9.5 | -4.7 | -17.1 | **842.8** |
| Walker Run Backward | 363.8 | 246.0 | 229.4 | 167.4 | **366.0** |
| Cheetah Run | 843.7 | 338.1 | **621.4** | 251.2 | 543.8 |
| Cheetah Run Front | 473.8 | 202.4 | 143.1 | 108.4 | **317.6** |
| Cheetah Run Back | 657.4 | 294.4 | 407.6 | 171.2 | **462.3** |
| Cheetah Run Backwards | 693.8 | 384.3 | **626.6** | 335.6 | 521.6 |
| Cheetah Jump | 597.0 | 535.6 | 200.8 | 251.8 | **614.2** |
| Quadruped Walk | 369.3 | 258.1 | 145.2 | 76.5 | **855.6** |
| Quadruped Stand | 746.0 | 442.2 | 227.2 | 318.9 | **941.4** |
| Quadruped Run | 328.1 | 296.5 | 183.0 | 102.8 | **766.9** |
| Quadruped Jump | 689.6 | 478.3 | 168.3 | 190.5 | **820.2** |
| Quadruped Roll | 663.9 | 446.0 | 207.9 | 126.2 | **948.0** |
| Quadruped Roll Fast | 508.8 | 366.9 | 124.8 | 164.7 | **758.9** |
| **Mean** | 541.81 | 372.23 | 320.86 | 226.49 | **617.73** |
| **Medium** | 606.8 | 318.70 | 204.35 | 166.05 | **733.3** |

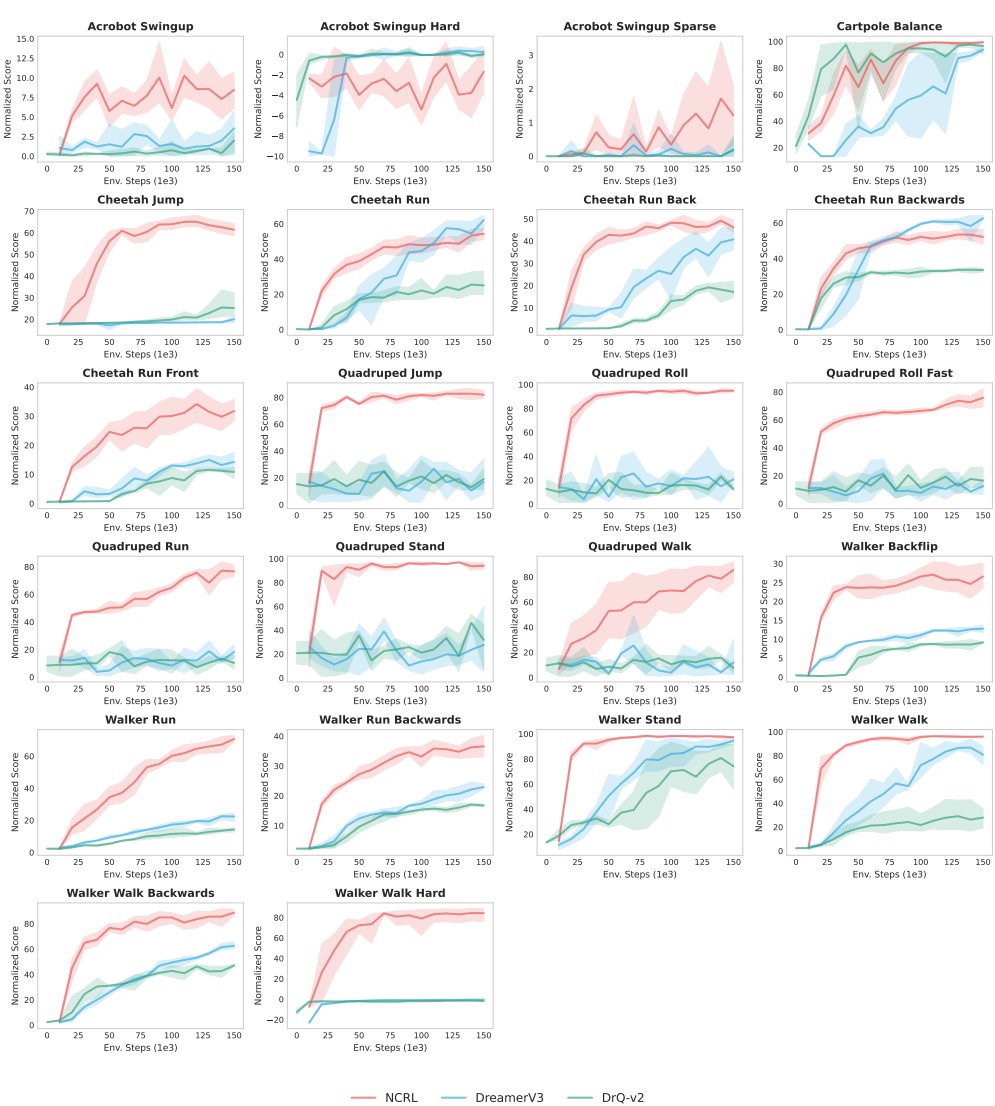

Figure 19: DMControl results. We report 5 seeds for NCRL and 3 seeds for DrQ-v2 and DreamerV3.

## J  HYPERPARAMETERS

In this section, we list important hyperparameters used in NCRL.

Table 6: Hyperparameters used in NCRL.

| Hyperparameter | Value |
|---|---|
| **Pre-training** | |
| Stacked images | 1 |
| Pretrain steps | 200,000 |
| Batch size | 16 |
| Sequence length | 64 |
| Replay buffer capacity | Unlimited |
| Replay sampling strategy | Uniform |
| RSSM | |
|   Hidden dimension | 12288 |
|   Deterministic dimension | 1536 |
|   Stochastic dimension | 32 * 96 |
|   Block number | 8 |
|   Layer Norm | True |
| CNN channels | [96, 192, 384, 768] |
| Activation function | SiLU |
| Optimizer | |
|   Optimizer | Adam |
|   Learning rate | 1e-4 |
|   Weight decay | 1e-6 |
|   Eps | 1e-5 |
|   Gradient clip | 100 |
| **Fine-tuning** | |
| Warm-up frames | 15000 |
| Execution Guidance Schedule | linear(1,0,50000) for DMControl |
| | linear(1,0,1,150000) for Meta-Wolrd |
| Action repeat | 2 |
| Offline data mix ratio | 0.25 |
| Discount | 0.99 |
| Discount lambda | 0.95 |
| MLPs | [512, 512, 512] |
| MLPs activation | SiLU |
| Actor critic learning rate | 8e-5 |
| Actor entropy coef | 1e-4 |
| Target critic update fraction | 0.02 |
| Imagine horizon | 16 |

# K    TASK VISUALIZATION

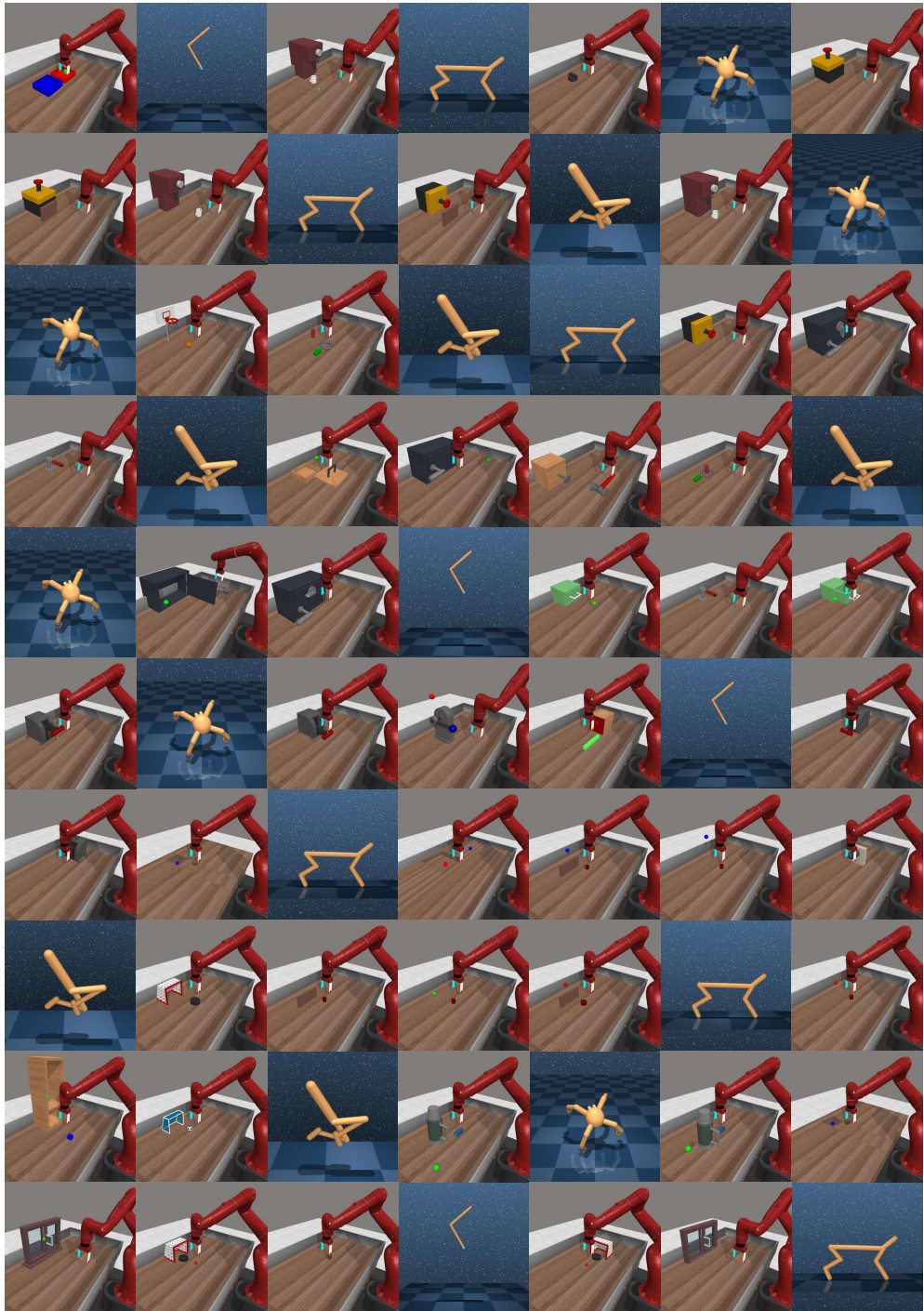

Figure 20: Visualization of tasks from DMControl and Meta-World used in our paper.

