# OpenReview forum: "Efficient Reinforcement Learning by Guiding World Models with Non-Curated Data"
_ICLR.cc/2026/Conference — ICLR 2026 Poster_

### Official Review · Reviewer_dmPf · 2025-10-27

**Soundness:** 3
**Presentation:** 3
**Contribution:** 3
**Rating:** 6
**Confidence:** 4

**Summary:**

This paper proposes a two-stage RL from offline data methods, which can leverage mixed quality offline data without reward labels. A world model is trained on these unlabeled offline data at stage one. At stage two, offline data relevant to online tasks, measured by the neural feature distance between online samples and offline trajectories, is retrieved to enrich the initial state distribution sampling. A behavior cloning policy $\pi_\text{bc}$ is pretrained on offline data to alternate sampling with the target policy during online training for enriching exploration.

Comprehensive experiments on locovisual tasks were conducted to verify the empirical advantages of this approach, compared to baseline RL methods that use offline data, as well as RL methods that learn from scratch.

**Strengths:**

Pros:

\+ This paper tackles an intriguing question of how to utilize imperfect, unlabeled, yet largely available offline data to boost RL performance, which is of practical importance.

\+ Technically solid methods that were clearly presented.

\+ Strong empirical performance with a gain of large margins, compared with baseline methods.

\+ Comprehensive evaluations on a wide range of baseline methods and locomotion tasks

**Weaknesses:**

Cons:

\- Vague descriptions about related work, such as "uses a series of complex techniques to stabilize training" (line 129) or "or carefully curated dataset" (line 131) - authors could've briefly summarized what techniques were used, and how curated data is generated in these related works.

\- Unclear definition of mixed quality data. Figure 9 alone does not sufficiently indicate the data quality, but only shows that imitation learning on these data leads to limited performance. See my question below for a follow-up.

**Questions:**

- Has the scaling world model to 280M parameters also been applied to baseline methods for fair comparison?
- How does Figure 2 support this: "The world model’s accuracy degrades on states visited by the early-stage policy, especially when the offline data distribution is narrow, hurting sample efficiency." (Line 213-214)
- Does training a bc policy on offline data rely on an unmentioned prerequisite: the mix quality data is not poor data? As mentioned, "offline data often contains valuable information like near-expert trajectories and diverse state-action coverage" (line 260). In my intuition, if the offline data quality is low, leveraging a bc policy would not help much in this algorithm. Have the authors explored to what extent the quantity and quality of offline data impact the algorithm's performance?

---

> ### Author Response · Authors · 2025-11-21
>
> We thank Reviewer dmPF for the constructive comments and for recognizing that our work tackles an intriguing research question, is technically solid, and demonstrates strong empirical performance with comprehensive evaluation. Your feedback has directly helped us clarify our assumptions, refine the related work discussion, and strengthen the empirical analysis of data quality and model scaling in the revised manuscript.
>
> ---
> > **W1:** Vague descriptions about related work, such as "uses a series of complex techniques to stabilize training" (line 129) or "or carefully curated dataset" (line 131) - authors could've briefly summarized what techniques were used, and how curated data is generated in these related works.
>
> **AW1:**
> We thank the reviewer for the comment. We have modified the related work to add more details, highlighted with blue color.
>
> The mentioned vague descriptions are originally detailed in the extended related work in Appendix C.
> Specifically, for the mentioned descriptions:
> "uses a series of complex techniques to stabilize training (line 129)" is detailed as:
> >"MOTO (Rafailov et al., 2023) proposes a model-based offline-to-online RL method with reward-labeled data, and requires model-based value expansion, policy regularization, and controlling epistemic uncertainty, while our method leverages reward-free and multi-embodiment data and requires none of the techniques proposed by MOTO".
>
> "carefully curated dataset (line 131)" is detailed as:
> >"RLPD (Ball et al., 2023) is proposed and demonstrates strong performance by simply concatenating offline and online data, but requires reward-labeled task-specific offline data and does not address multi-embodiment scenarios. ExPLORe (Li et al., 2023) labels reward-free offline data using approximated upper confidence bounds (UCB) to solve hard exploration tasks, but relies on near-expert data for the target tasks, while we consider a more general setting with non-curated data."
>
> We have modified the related work section to include the details in the revised paper.
>
> ---
>
> > **W2:** Unclear definition of mixed quality data. Figure 9 alone does not sufficiently indicate the data quality, but only shows that imitation learning on these data leads to limited performance. See my question below for a follow-up.
>
> We appreciate the reviewer’s request for a more precise definition of “mixed-quality” data.
>
> For DMControl, we use 10k trajectories across 5 embodiments, collected by unsupervised RL agents. These trajectories vary in competence and coverage. As the unsupervised RL agents are trained to maximize "internal" rewards rather than specific tasks, the dataset for DMControl does not contain expert trajectories for a specific task (e.g., Walk, Run, etc.)
>
> For Meta-World, we collect data using TDMPC-v2 agents with injected Gaussian noise ($\sigma \in {0.1, 0.3, 0.5, 1.0, 2.0}$), which intentionally corrupts the policies and produces trajectories of varying success and quality. In practice, such a mixture of successful, partially successful, and failed behaviors can naturally arise from, for instance, noisy or partial human demonstrations collected through teleoperation. Together with the new ablation discussed in Q3 (Appendix A.4), we hope these have led to a clearer description of the data regime we consider.

---

> > ### Author Response · Authors · 2025-11-21
> >
> > ## Questions:
> >
> > > **Q1:** Has the scaling world model to 280M parameters also been applied to baseline methods for fair comparison?
> >
> > **AQ1:** We thank the reviewer for this valuable observation. For the Dreamer baseline, we adopt a commonly used model configuration, which has a smaller model size. In RL, increasing model size does not necessarily lead to better performance; the relationship is often non-monotonic.
> > Indeed, although DreamerV3 includes a model-size ablation, it still uses a small model for DMControl tasks [1].
> > To empirically evaluate the impact of Dreamer’s model size, we compared DreamerV3 with a variant that matches the model size used in NCRL, as shown in Fig. 15 of the revised paper. We found that increasing Dreamer’s model size leads to mixed results: performance improves on Walker Run but degrades on Quadruped Walk. Nonetheless, NCRL consistently outperforms both DreamerV3 variants.
> > We have included the discussion in Appendix A.5 of the revised paper.
> >
> >
> > >**Q2:** How does Figure 2 support this: "The world model’s accuracy degrades on states visited by the early-stage policy, especially when the offline data distribution is narrow, hurting sample efficiency." (Line 213-214)
> >
> > **AQ2:** We thank the review for pointing this out. We revise this sentence as:
> > "World model accuracy can degrade if a significant distributional shift exists between the offline and online data. This degradation is particularly pronounced when the offline data distribution is narrow, which may create a substantial state-space gap."
> >
> >
> > >**Q3:** Does training a bc policy on offline data rely on an unmentioned prerequisite: the mix quality data is not poor data? As mentioned, "offline data often contains valuable information like near-expert trajectories and diverse state-action coverage" (line 260). In my intuition, if the offline data quality is low, leveraging a bc policy would not help much in this algorithm. Have the authors explored to what extent the quantity and quality of offline data impact the algorithm's performance?
> >
> >
> > **AW3:** We thank the reviewer for the valuable question.
> >
> > First, we agree that any method leveraging offline data implicitly assumes that the data contains some useful signal. The distinction lies in how strong these assumptions are. For example, standard offline RL methods typically require reward-labeled, task-specific datasets [1,2], and methods such as [3,4] assume access to expert demonstrations. In contrast, our method relaxes these requirements by enabling the agent to benefit from reward-free, mixed-quality, and multi-embodiment data.
> >
> > Second, following your suggestion, we conducted an ablation to analyze how retrieved offline data affects performance. For three challenging Meta-World manipulation tasks, we progressively replaced the retrieved task-relevant trajectories with 0%, 25%, 50%, 75%, and 100% trajectories that lie far from the target task in the latent space. As reported in the revised appendix (Appendix A.4), NCRL remains robust even when a substantial fraction of the retrieved data is task-irrelevant. As expected, performance declines only when nearly all retrieved trajectories are far from the target task.
> >
> >
> > ---
> > [1] Fujimoto, Scott, and Shixiang Shane Gu. "A minimalist approach to offline reinforcement learning." Advances in neural information processing systems 34 (2021): 20132-20145.
> >
> > [2] Kumar, Aviral, et al. "Conservative q-learning for offline reinforcement learning." Advances in neural information processing systems 33 (2020): 1179-1191.
> >
> > [3] Hansen, Nicklas, et al. "Modem: Accelerating visual model-based reinforcement learning with demonstrations." arXiv preprint arX
> >
> > [4] Lancaster, Patrick, et al. "Modem-v2: Visuo-motor world models for real-world robot manipulation." 2024 IEEE International Conference on Robotics and Automation (ICRA). IEEE, 2024.

---

> > > ### Comment · Reviewer_dmPf · 2025-11-25
> > > **Response**
> > >
> > > Thanks to the authors for their detailed response and experimental results. I think they well addressed my comments. I would also vote for acceptance.

---

### Official Review · Reviewer_Uq2c · 2025-10-30

**Soundness:** 4
**Presentation:** 4
**Contribution:** 4
**Rating:** 10
**Confidence:** 4

**Summary:**

This paper investigates the question of how to improve sample efficiency of RL by using offline data. The specific situation this paper looks into is the case where there is a large amount of offline data but that data is unstructured and not annotated. The paper pre-trains a world model on such data, consisting of many embodiments and then uses fine tuning to optimize for a specific task. The novel methodology of this work is the enabling of fine tuning of the world model to optimize for specific tasks and improve performance. In the first stage they pre-training their world model on offline image-based data. In the fine tuning stage their agent interacts with the environment and collects data for world model training. Their results show that their pre-training is able to improve sample efficiency greatly and doesn't hurt final performance.

**Strengths:**

Great motivating question as well as a practical method.

Very strong results using a single world model for multiple tasks. This is a difficult challenge and this paper provides impressive empirical results.

Paper is well written and very clear. Method is described well and the paper is easy to read. Empirical results are displayed clearly.

Empirical results are impressive and seem to be statistically well done with confidence intervals and such.

**Weaknesses:**

I do not have any strong weaknesses in this paper that I can find.

**Questions:**

Around line 266 you describe using a behavior cloned policy to choose actions. Does this cause any issues with the data now being off policy? I'm not sure how the dreamer architecture handles off policy data (it might handle it completely fine) but I'm wondering if this was something you considered?

The central problem you identify is distribution shift between the pre train data and the fine tune data. I see why this could cause problems but would your method prevent or slow down (in some settings) convergence of the world model to the best points in the world space? I'm just thinking that keeping around old data is fine and prevents a distribution shift but what about in the cases where the distribution shift is actually required. Possible example would be long horizon tasks where we only have pre train data for a small portion of the beginning, in this case we might not learn as well the longer ends of the task.

For the results (280) did you increase the size of the dreamer world model as well? It is unclear to me if this would make any difference or if it might even make dreamer perform worse but I missed if you described this somewhere.

---

> ### Author Response · Authors · 2025-11-21
>
> We sincerely thank reviewer Uq2c for recognizing the strengths of our work and for providing valuable comments to help improve the paper. We address each question raised in the review below.
>
> > **Q1:** Around line 266 you describe using a behavior cloned policy to choose actions. Does this cause any issues with the data now being off policy?
>
> **AQ1:** We thank the reviewer for the observation. In Dreamer, the data collected through interaction with the environment is used to train the world model via supervised learning, while the policy and value functions are trained with “on-policy’’ data generated by unrolling the world model. From an algorithmic perspective, incorporating additional off-policy data should therefore be acceptable, as Dreamer naturally supports the use of off-policy data.
>
> Furthermore, unlike offline model-based RL where the agent is not allowed to interact with the environment, our setting still permits environment interaction, and the guidance policy is executed only according to a predefined schedule that decays over time.
>
> In our experiments, we did not observe clear evidence that execution guidance hinders training.
> We are happy to answer further questions raised during the discussion.
>
>
> > **Q2:** The central problem you identify is distribution shift between the pre train data and the fine tune data. I see why this could cause problems but would your method prevent or slow down (in some settings) convergence of the world model to the best points in the world space? I'm just thinking that keeping around old data is fine and prevents a distribution shift but what about in the cases where the distribution shift is actually required. Possible example would be long horizon tasks where we only have pre train data for a small portion of the beginning, in this case we might not learn as well the longer ends of the task.
>
> **AQ2:** We thank the reviewer for raising the important point regarding the need for distribution shift.
>
> We wish to clarify that our method is not designed to hinder the agent's exploration. The impact of the guidance policy is governed by a decaying schedule: as the training progresses, the probability of utilizing the guidance policy gradually diminishes. Simultaneously, as the agent collects new data, the growing online replay buffer naturally reduces the relative influence of the offline data distribution.
> Thus, the agent retains the necessary capacity to explore the environment and move away from the original offline data distribution as required.
> Intuitively, our method provides a *smooth transition* between offline pre-training and online fine-tuning, rather than imposing strict behavioral constraints typically found in standard offline RL approaches.
>
>
> > **Q3:** For the results (280) did you increase the size of the dreamer world model as well? It is unclear to me if this would make any difference or if it might even make dreamer perform worse but I missed if you described this somewhere.
>
>
> **AQ3:** We thank the reviewer for this valuable observation. For the Dreamer baseline, we adopt a commonly used model configuration, which has a smaller model size.
> Indeed, although DreamerV3 includes a model-size ablation, it still uses a small model for DMControl tasks [1].
> To empirically evaluate the impact of Dreamer’s model size, we compared DreamerV3 with a variant that matches the model size used in NCRL, as shown in Fig. 15 of the revised paper. We found that increasing Dreamer’s model size leads to mixed results: performance improves on Walker Run but degrades on Quadruped Walk. Nonetheless, NCRL consistently outperforms both DreamerV3 variants.
> We have included the discussion in Appendix A.5 of the revised paper.
>
> ---
> [1] Hafner, D., Pasukonis, J., Ba, J. et al. Mastering diverse control tasks through world models. Nature 640, 647–653 (2025). https://doi.org/10.1038/s41586-025-08744-2

---

### Official Review · Reviewer_LFwp · 2025-11-01

**Soundness:** 3
**Presentation:** 3
**Contribution:** 2
**Rating:** 8
**Confidence:** 3

**Summary:**

The paper introduces NCRL (Non-Curated offline data for efficient RL), an algorithm that leverages non-curated, reward-free, mixed-quality, and multi-embodiment datasets to improve sample efficiency in offline-to-online reinforcement learning. NCRL first pretrains a multi-task, multi-embodiment world model on this data, then reuses the same offline trajectories during fine-tuning through two mechanisms:
(1) experience rehearsal, which retrieves task-relevant trajectories to mitigate distribution shift, and
(2) execution guidance, a behavior-cloned prior policy that intermittently steers the agent during online training.
In  Meta-World and DMControl tasks, NCRL outperforms baselines and matches the performance of models trained with 3.3–6.7× larger budgets. It also surpasses several offline-data and world-model pretraining baselines.

**Strengths:**

1. **Clarity and motivation.** The paper is well-written. The contributions are clearly articulated, the relationship to prior work is carefully established, and the distinctions from existing approaches are explicit and well-justified. I especially appreciate that the authors explain the reasoning behind key design choices throughout. Figure 2 provides a particularly clear and compelling motivation for the proposed approach.
1. **Experimental design** The experiments are thoughtfully designed and demonstrate consistent, meaningful improvements of NCRL over relevant baselines. The ablations are comprehensive and provide useful information (e.g. showing that both core components of NCRL contribute to performance gains)
1. **Reproducibility.** The paper includes extensive implementation and hyperparameter details, making the method straightforward to reproduce and encouraging further study or application.

**Weaknesses:**

I vote to accept, though I have a few minor comments and clarifying questions:

1. At the end of Section 3, the paper refers to theoretical results in Appendix B, but the main text does not summarize what these results show or why they are relevant. It would help readers if the paper briefly explained, in the main body, what guarantees or insights the theory provides (e.g., how experience rehearsal or execution guidance theoretically improves performance or mitigates distribution shift), etc.
1.  In Tables 3 and 4, NCRL solve most Meta-World tasks with high probability but not all of them. Could the authors clarify what makes these tasks challenging? Are these differences due to the inherent difficulty of the tasks, the limited training budget of 150k steps, or particular weaknesses in retrieval or guidance?

Typos:
1. “reuses the this data” --> “reuses this data.”
2. “Comparsion” --> “Comparison” in Fig. 11 caption
4. "reutrn" --> "return" in code snippet

**Questions:**

See weaknesses.

---

> ### Author Response · Authors · 2025-11-21
>
> We are glad that Reviewer LFwp also voteed to accept our manuscript. We would also like to take the time to thank you for the valuable feedback which has enabled us to improve the manuscript. We give our responses to each point raised in the review below.
>
> > **W1:** It would help readers if the paper briefly explained, in the main body, what guarantees or insights the theory provides (e.g., how experience rehearsal or execution guidance theoretically improves performance or mitigates distribution shift), etc.
>
> **AW1:**
>
> We thank the reviewer for the valuable suggestion. As recommended, although our paper is primarily empirical, we agree that briefly discussing the theoretical conclusions in the revised version would be beneficial.
>
> Specifically, in Proposition 1, we show that if the retrieved states are close to the online states, i.e., $||s_\text{retrieved} −s_\text{on}|| < \epsilon$ ensured by Eq (4), then experience retrieval reduces distribution shift during online fine-tuning compared to using the full offline dataset directly. This result supports the use of experience rehearsal.
>
> In Proposition 2, we show that experience retrieval helps mitigate catastrophic forgetting during online fine-tuning by acting as a regularizer that preserves performance on the retrieved offline data distribution.
>
> In Proposition 3, we show that at the beginning of training, the mixture policy can improve the performance by a lower bound compared to the RL policy without using the execution guidance.
>
>
> We have integrated these conclusions into the main body of the revised paper.
>
>
> > **W2:** In Tables 3 and 4, NCRL solve most Meta-World tasks with high probability but not all of them. Could the authors clarify what makes these tasks challenging?
>
> **AW2:**
> We selected the tasks that exhibited low performance at 150k environment steps and reran them with a larger training budget. The results are provided in Appendix A.6 of the revised paper. As shown, most tasks achieve higher success rates as the training budget increases. We also observed that many of these challenging tasks involve very small objects of interest (e.g., the objects of interest only take a few pixels in Basketball and Pick Place), relatively strict success criteria, or long-horizon behavior (e.g., Assembly and Stick Pull).
>
>
> > Typos
>
> We thank the reviewer for taking the time to read our manuscript carefully. We have fixed the identified typos and will have an additional thorough pass for the camera-ready version.

---

> > ### Comment · Reviewer_LFwp · 2025-11-25
> >
> > Thank you for the response! I appreciate the revisions you've made. I maintain my score and have no further comments.

---

### Official Review · Reviewer_RREe · 2025-11-01

**Soundness:** 4
**Presentation:** 4
**Contribution:** 3
**Rating:** 8
**Confidence:** 3

**Summary:**

The work addresses the challenge of improving sample efficiency in reinforcement learning (RL) by leveraging non-curated offline datasets which are reward-free, of mixed quality, and collected from multiple embodiments during online reinforcement learning. They investigate this in the world modeling case, and observe that naively replaying this data fails because of distribution mismatch. To overcome this, the paper proposes NCRL (Non-Curated offline data for RL), a two-stage framework: first train a task-agnostic world model on the offline data, and then fine-tune it on the target task. Specifically, the authors introduce two techniques during the fine-tuning phase to take advantage of this data: (1) Experience Rehearsal, which retrieves task-relevant trajectories from the offline dataset (based on neural feature distance using the world model encoder) and replays it during fine-tuning, and (2) Execution Guidance via Prior Actors, which randomly alternates between a prior policy trained on the retrieved offline data and the online RL policy, improving exploration.

Experiments demonstrate that in the low-sample regime, NCRL improves significantly over learning-from-scratch baselines and match their performance in the high-sample regime; it also help sin a continual learning scenario with a sequence of tasks. Careful ablation experiments show each component's contribution to the observed improvement and improvement over directly using uncertainty-aware rewards for the offline data.

**Strengths:**

The paper is written with clarity; the exposition of methods and step-by-step motivation is reasonable / comprehensive. In particular, the high-level motivation for the setting is very clear.

The experimental methodology is thorough and high-quality: covers a wide range of environments, uses appropriate metrics (normalized scores, learning curves with confidence intervals), and includes ablations to justify the algorithm’s design.

The novelty is primarily in framing the setting of a mixed-quality offline dataset; the methods fall quite naturally from this, and has related work with different assumptions (e.g. JSRL).

**Weaknesses:**

While the high-level motivation of using mixed-quality offline dataset makes perfect sense, the details of the way the offline dataset is constructed is somewhat artificial (e.g. agents corrupted with noise) and has characteristics which naturally lends itself to the proposed method; specifically, it's likely to cover similar state-space as some of the tasks, which means smart retrieval methods from this dataset would naturally help in the low-data regime. Similarly, a prior policy trained on retrieved tasks would help exploration simply by virtue of having seen similar tasks before. Perhaps putting stronger explanation on why this kind of dataset is reasonable to assume / arises naturally (e.g. in real-world robotics settings) would better motivate the method.

The impact of showing improvements in the low-data regime with these particular simulation settings is less clear, given cheapness of simulation data and the asymptote to the same performance with more environment steps.

**Questions:**

This line of work [1], [2] are relevant given the emphasis on using demonstrations in a world-modeling approach.

Further analysis of exactly how the offline dataset matches the current task distribution would be helpful, i.e. could we quantify or visualize how much of the improvement comes from retrieving trajectories matching the current task state-space? Does the fact that the dataset is cross-embodiment matter if all the retrieved trajectories are the same embodiment?

Any further analysis on the ablations in Figure 6? Specifically, it seems that execution guidance is the main differentiator for achieving nonzero reward for some subset of the Meta-World tasks.

[1] Hansen, Nicklas, et al. "Modem: Accelerating visual model-based reinforcement learning with demonstrations." arXiv preprint arX
[2] Lancaster, Patrick, et al. "Modem-v2: Visuo-motor world models for real-world robot manipulation." 2024 IEEE International Conference on Robotics and Automation (ICRA). IEEE, 2024.

---

> ### Author Response · Authors · 2025-11-21
>
> We are pleased that Reviewer RREe is inclined to accept our paper and appreciates that it is clearly written, well-motivated, and supported by thorough and high-quality experiments. We also thank the reviewer for their constructive comments, which have helped us further improve the manuscript. We address each point below.
>
> ---
> > **W1** Perhaps putting stronger explanation on why this kind of dataset is reasonable to assume / arises naturally (e.g. in real-world robotics settings) would better motivate the method.
>
> **AW1:**
> We thank the reviewer for the valuable comment.
> In real-world settings, such useful yet imperfect data naturally arises from, for instance, noisy or partial human demonstrations collected via teleoperation. This leads to a mixture of successful, partially successful, and failed behaviors.
> Also, an offline dataset consisting of robots' prior experience across related tasks could be helpful for the training, but labeling the offline data with rewards in a retrospective way could be expensive.
> Our method relaxes the strong assumptions made in prior work [1,2,3,4], which typically require reward-labeled [3,4] or expert-quality datasets [1,2]. Instead, we show that an RL agent can effectively leverage multi-embodiment, mixed-quality, and reward-free offline data, which is a more relaxed and reasonable assumption of the offline data.
>
> > **W2** The impact of showing improvements in the low-data regime with these particular simulation settings is less clear, given cheapness of simulation data and the asymptote to the same performance with more environment steps.
>
> **AW2:**
> We thank the reviewer for the comment. In the RL setting, it is common to use simulators to evaluate sample efficiency [6]. Our method shows strong potential for enabling practical real-world RL. For example, prior methods require more than 1M samples to learn locomotion with the Quadruped robot in DMControl, whereas our approach succeeds with only 100 trajectories. This significant reduction in data requirements could make real-world RL feasible.
>
> Regarding the asymptotic performance, our method achieves higher final performance than training-from-scratch baselines on several tasks, including Assembly from MetaWorld and Cheetah Run Hard from DMControl etc., showing the benefit of leveraging offline data for achieving better asymptotic performance.
>
> ---
>
> ## Questions
>
> > **Q1** This line of work [1], [2] are relevant given the emphasis on using demonstrations in a world-modeling approach.
>
> **AQ1:** We thank the reviewer for pointing out these relevant works. We have included them in the related work section. Our approach differs from these methods mainly in the assumptions about the offline dataset. While [1][2] assume access to expert demonstration data, our method relaxes this requirement and is designed to utilize non-curated offline data. This difference in data assumptions naturally leads to methodological differences from these prior works.

---

> ### Author Response · Authors · 2025-11-21
>
> > **Q2:** Further analysis of exactly how the offline dataset matches the current task distribution would be helpful, i.e. could we quantify or visualize how much of the improvement comes from retrieving trajectories matching the current task state-space? Does the fact that the dataset is cross-embodiment matter if all the retrieved trajectories are the same embodiment?
>
> **AQ2:** We thank the reviewer for the suggestion. We conducted an additional ablation study to evaluate the impact of retrieved trajectories that do not match the target task. Specifically, for three challenging manipulation tasks from MetaWorld, we progressively replaced the retrieved task-relevant trajectories with 0%, 25%, 50%, 75%, and 100% trajectories that lie far from the target task in the latent space. As shown in the revised paper, Appendix A.4, our method remains robust even when a portion of the retrieved offline data is task-irrelevant.
>
> One benefit of our method’s ability to utilize multi-embodiment data is that it removes the need to pre-train separate world models for each embodiment. Another potential advantage worth exploring in future work is the generalization ability across embodiments [5].
>
>
> > **Q3:** Any further analysis on the ablations in Figure 6? Specifically, it seems that execution guidance is the main differentiator for achieving nonzero reward for some subset of the Meta-World tasks.
>
> **AQ3:** We thank the reviewer for the observation. To apply execution guidance, the retrieval step is required as a prerequisite for learning the guidance policy. In our setting, the offline dataset is large and contains a substantial amount of task-irrelevant data, making it infeasible to learn the guidance policy directly from the full dataset. The retrieval step enables us to efficiently select task-relevant data, thereby simplifying and stabilizing the training process.
> Beyond facilitating the learning of the guidance policy, we also found that the rehearsal mechanism helps stabilize performance on several tasks, including Walker Run and Cheetah Run Hard. We are happy to conduct additional experiments further raised during the discussion.
>
>
> ---
> [1] Hansen, Nicklas, et al. "Modem: Accelerating visual model-based reinforcement learning with demonstrations." arXiv preprint arX
>
> [2] Lancaster, Patrick, et al. "Modem-v2: Visuo-motor world models for real-world robot manipulation." 2024 IEEE International Conference on Robotics and Automation (ICRA). IEEE, 2024.
>
> [3] Fujimoto, Scott, and Shixiang Shane Gu. "A minimalist approach to offline reinforcement learning." Advances in neural information processing systems 34 (2021): 20132-20145.
>
> [4] Kumar, Aviral, et al. "Conservative q-learning for offline reinforcement learning." Advances in neural information processing systems 33 (2020): 1179-1191.
>
> [5] Ai, Bo, et al. "Towards Embodiment Scaling Laws in Robot Locomotion." Coference on Robot Learning, 2025.
>
> [6] Ball, Philip J., et al. "Efficient online reinforcement learning with offline data." International Conference on Machine Learning. PMLR, 2023.

---

### Author Response · Authors · 2025-11-21
**[Updated] General Response by Authors**

We sincerely thank all reviewers RREe, LFwp, Uq2c, dmPf, and ACs for the time they dedicated to reviewing our work. We are grateful for the comments, constructive suggestions, and detailed questions, all of which have directly contributed to strengthening and clarifying our manuscript. We truly view this process as a collaborative effort, and we appreciate the reviewers’ engagement with both the high-level motivation and the finer technical details of our approach.

---
Across the reviews, we are glad to receive several *consistent points of positive feedback*:
- Clarity and motivation: RREe, LFwp, and Uq2c praised the clarity and motivation.
- Strong and comprehensive experimental evaluation: RREe, LFwp, and dmPf each highlighted the breadth and quality of our experiments and ablations, with Uq2c describing the empirical results as “impressive.”
- Practical relevance and contribution: dmPf underscored the practical importance of leveraging imperfect offline data, while Uq2c emphasized the significance of successfully using a single world model across many tasks.
- Overall assessment: We are encouraged that RREe and LFwp recommended acceptance, and Uq2c recommended strong acceptance. **During the rebuttal, reviewer dmPf also recommended acceptance (score 8)**.

---

## Summary of modifications
**We are glad that during the rebuttal, reviewers LFwp and dmPf indicated that their concerns were properly addressed. And reviewer Uq2c indicated that no further weaknesses in the initial review.**
During the rebuttal, the reviewers’ constructive suggestions helped us substantially improve the paper. In direct response to their feedback, we have:
- Refined the related-work discussion to clarify assumptions and techniques (addressing dmPf).
- Included new ablations on retrieval relevance and robustness to low-quality offline data (addressing RREe and dmPf).
- Clarified model-size considerations (addressing Uq2c and dmPf).
- Added a succinct summary of the theoretical insights to the main text (addressing LFwp).
- Provided a clearer characterization of mixed-quality data (addressing dmPf).
- Added new training results for challenging MetaWorld tasks with a larger training budget (addressing LFwp).

We believe these changes have strengthened the paper, and we appreciate the reviewers’ guidance in arriving at a clearer, more rigorous, and more thoroughly justified presentation.

Once again, we thank all reviewers for their encouragement and help in improving the manuscript.

---

### Meta-Review · Area_Chair_3bTK · 2026-01-05

**Summary:**

This paper proposes a two-stage RL framework that leverages mixed-quality offline data without reward labels, using a large world model to improve the sample efficiency of online RL. In the first stage, a world model is pre-trained on the offline data. In the second stage, the pretrained world model, offline data, and online interactions are jointly used to perform RL-based fine-tuning. The main contribution lies in the simple procedural design, which leads to performance gains demonstrated through comprehensive experiments. These include the two-stage training strategy, selective reuse of offline data during fine-tuning, and the use of a prior policy, while the world model itself is adopted from existing literature.

Extensive experiments on locovisual tasks from Meta-World and DMControl validate the empirical advantages of the proposed approach over baseline RL methods that utilize offline data, as well as methods that learn from scratch. The paper received strong reviewer scores (8, 8, 10, 6). Reviewers consistently found the paper to be well written and to address an intriguing and timely problem. The experimental evaluation is viewed as comprehensive and well designed, demonstrating consistent and impressive performance gains across multiple tasks. The reviewers’ concerns were generally addressed satisfactorily in the authors’ rebuttal.

**Reviewer Concerns:**

All reviewers' concerns were addressed. Reviewer dmPf and LFwp acknowledged the authors' responses.

**Reviewer Scores:**

The scores were originally pretty high on this paper.

---

### Decision · Program_Chairs · 2026-01-26

Accept (Poster)